# Biogenic sunflower oil-chitosan decorated fly ash nanocomposite film using white shrimp shell waste: Antibacterial and immunomodulatory potential

Seham S. Alterary[1]*, Musarat Amina[2]*, Maha F. El-Tohamy[1]

1 Department of Chemistry, College of Science, King Saud University, Riyadh, Saudi Arabia, 2 Department of Pharmacognosy, Pharmacy College, King Saud University, Riyadh, Saudi Arabia

* salterary@ksu.edu.sa (SSA); mamina@kau.edu.sa (MA)

**Data Availability Statement:** All relevant data are within the paper and its supporting information.

## Abstract

A new sunflower oil-chitosan decorated fly ash (sunflower oil/FA-CSNPs) bionanocomposite film was synthesized using the extract of *Litopenaeus vannamei* (White shrimp) and evaluated as an antibacterial and immunomodulatory agent. Fly ash-chitosan nanoparticles were produced by using chitosan (CS) isolated from white shrimp extract, glacial acetic acid and sodium tripolyphosphate solution as cross-linkage. The ultrafine polymeric sunflower oil-CS film was fabricated by treating fly ash-chitosan nanoparticles with sunflower oil in glacial acetic acid under continuous stirring for 24 h. The nanostructure of the fabricated polymeric film was confirmed and characterized by different microscopic and spectroscopic approaches. The surface morphology of pre-synthesized bionanocomposite film was found to be homogenous, even and without cracks and pores. The crystallinity of formed bionanocomposite film was noticed at angles ($2\theta$) at 12.65˚, 15.21˚, 19.04˚, 23.26˚, 34.82˚, and 37.23˚ in the XRD spectrum. The fabricated film displayed excellent stability up to 380 ˚C. The formed sunflower oil/FA-CSNPs bionanocomposite film showed promising antibacterial towards *Bacillus subtilis* with highest zone of inhibition of 34 mm and *Pseudomonas aeruginosa* with zone of inhibition of 28 nm. The as-synthesized bionanocomposite film exhibited highest cell viability effect (98.95%), followed by FA-CSNPs (83.25%) at 200 µg mL$^{-1}$ concentrations. The bionanocomposite film exerted notable immunomodulatory effect by promoting phagocytosis and enhancing the production of cytokines (NO, IL-6, IL-1β, and TNF-α) in macrophage-derived RAW264.7 cell line.

## Introduction

Nanobiotechnology prudently links the efficiency of biological materials and the tools to develop minute synthetic structures [1]. These nanoscale structures can effectively be incorporated into biomedical applications to develop highly functional biomolecules for diagnosis, drug delivery systems, immunomodulatory and antibacterial biosensors, and films [2, 3]. The

**Funding:** The authors extend their appreciation to the Deputyship for Research & Innovation, Ministry of Education in Saudi Arabia for funding this research work through the project no. (IFKSURG-1588). This research was funded by Researchers Supporting Project in King Saud University and the code number is (RSP-2021/195). The funder has full participation in the manuscript beside financial support. The authors participated in visualization, formal analysis, and validation.

**Competing interests:** The authors have declared that no competing interests exist.

benefits of sustainable nanocomposite by mixing nanostructures with large surface area and lower dimensional polymeric layers resulted in various useful applications [4]. The smart use of environmentally friendly synthetic nanocomposites has huge potential for reducing the expanding negative effects of contemporary technology on the environment, such as pollution and waste disposal [5].

The creation of nanoparticles by combining synthetic biomaterials with biological molecules based on substances obtained from natural resources is a common practice for many biological techniques [6]. Recently, the control of organic waste disposal is a growing issue for the legislation and environmental safety. Every year, massive amount of agricultural and industrial wastes has been generated and leads to many disposals, governance and environmental problems [7–10]. Reutilization of agricultural and industrial trash into value-added products is an important approach for long-term manner to attain sustainable cycles in the industry as well as in water waste treatments [11–14]. This adheres to the concepts of minimum waste generation and waste-to-wealth mission. In addition to synthetic organic components of solid waste disposal from the municipality, unwanted fruit and vegetable peels, flyash, egg shells and shrimp peels can all be utilized in the field of nanotechnology [15]. The agricultural and industrial trash, particularly ashes, are considered as essential raw materials and can be exploited as a starting point for the formation of useful nanomaterials. For instance, 75 to 120 million tons of flyash of fly ash is generated in various industrial processes by different countries annually. An enormous amount of flyash that is produced each year is handled carelessly and has negative environmental effects [16]. Thus, recycling of waste flyash has dual goals of reducing waste and recycling materials to create usable goods and protect the environment from negative effects that are generated by the accumulation of these wastes [17].

Fly ash (FA) is the finest coal combustion product made up of small inorganic mineral particles with a minor amount of carbon and has a crystalline appearance [18]. Thus, it's a cost-effective cement addition utilized for the stabilization of geotechnical engineering that requires loose soils. It has been applied in various agricultural and environmental applications [19]. Moreover, studies have reported that enwrapped ZnS nanoparticles with fly ash exerted good antibacterial potential and a promising photocatalytic activity for wastewater treatment, dye removal, and food packaging applications [20]. Several studies on the synthesis of nanoparticles using fly ash and its biological activities have been addressed in the literature [21]. Yadav et al. demonstrates a practical and affordable approach for producing silica nanoparticles from silico-aluminous class using fly ash [22]. Another study conducted by Mitiko and Denise reported the preparation of magnetic adsorbent by mixing coal fly ash with ziolite [23]. The production of amorphous silica nanoparticles from burned paddy straw has been accomplished chemically using fly ash [24]. However, there are only two reports mentioning chitosan loaded fly ash nanocomposite [25, 26].

Chitosan (CS) is an abundant prevalent biopolymer polysaccharide and is recognized for its efficient bio-decomposition after cellulose [27]. It has remarkable biocompatibility [28] used as a viral vector for gene therapy [29], and muco-adhesiveness [30]. It has been assigned as "safest polymer" by Food and Drug Administration United States [31]. Chitosan has shown many biomedical properties including anticancer, hemostasis, wound healing, anti-inflammatory, and antimicrobial [32–36]. Studies reported that casted films using chitosan and guar gum exhibited antimicrobial activities [37]. Silver and gold/chitosan nanocomposites films displayed inhibitory effects towards various bacterial strains [38]. Chitosan has also exerted excellent film layer abilities with an exceptional gas barrier, improved permeability of water vapor, protection from Ultra-violet light [39], and showed promising ability to encapsulate therapeutic drugs as well as release them in a regulated manner [40]. Moreover, its cationic feature facilitates effective polyelectrolyte interactions and ionic linkages with multivalent anions, which

makes it easy to derivatize or functionalize [41]. Different approaches have been applied to prepare chitosan nanostructures, such as nanoprecipitation, reverse micellizaton, covalent cross-linking, ionic gelation, and emulsion solvent evaporation [42–44]. Various interesting materials have been suggested to enhance the physiochemical and adsorptive properties of chitosan composites including zinc oxide [45], activated carbon [46], montmorillonite [47], and fly ash (FA) [48].

In the recent times, oil-dependent polymeric nanocomposite films have a wide range of uses, including, pharmaceutical packaging and biomedical equipment [49, 50]. These nano-particles exert viable and potent antimicrobial properties against various drug resistant microbes [51]. Considering, the therapeutic benefits of essential oils (EOs) to humans and the combination of these oils with nanoparticles as natural potential materials has opened new techniques to produce biomaterials for biomedical applications [52]. Thus, our study is focused on the preparation of oil-based fly ash bionanocomposite decorated with natural extracted chitosan nanoparticles. The bionanocomposite film formed will has synergetic effect due to the combination of components of oil and fly ash with enormous beneficial characteristics of chitosan nanoparticles that can be explored for promising biomedical applications. To the best of our knowledge this is the first report for preparation of bionanocomposite film using sunflower oil- FA-CSNPs.

Herein, a novel polymeric sunflower oil- FA-CSNPs bionanocomposite film was fabricated using chitosan nanoparticles in the presence of sunflower oil decorated with fly ash. *L. vanna-mei* shell extract was used for the production of chitosan nanoparticles. The developed film was characterized by UV-vis, FT-IR, XRD, and SEM analyses. The pre-synthesized bionano-composite film was evaluated for antibacterial potential against two bacterial strains *B. subtilis* (Gram-positive) and *P. aeruginosa* (Gram-negative). The formed film was also examined for the immunomodulatory activity against the macrophage-derived RAW264.7 cell line.

## Material and methods

### Chemicals and reagents

Hydrochloric acid (HCl, 37%), sodium hydroxide (NaOH, $\geq$ 97%), ethanol (95%), glacial ace-tic acid (CH₃COOH, 100%), sodium tripolyphosphate ($\geq$98%), Tween-80, acetone ($\geq$99.8%), Mueller-Hinton broth, Dulbecco's modified eagle medium (DMEM), fetal bovine serum (FBS), Tween 80 (10% low peroxide), osmium tetroxide (99.8%), 3-(4,5-dimethylthiazol-2-yl)-2,5-diphenyl-2H-tetrazolium bromide (MTT,98%), streptomycin sulfate, penicillin, 1-(4,5-dimethylthiazol-2-yl)-3,5-diphenylformazan, (Thiazolyl blue formazan crystals), dimethyl sulfoxide (DMSO, $\geq$ 99.9%), 3-Amino-7-dimethylamino-2-methylphenazine hydrochloride (neutral red, 0.09%), glutaraldehyde solution (50% in water), phosphate buffered saline (PBS, pH 7.4), Phagocytosis Assay kit (FITC-labeled *E. coli*), paraformaldehyde (95%), TRiPure^MT isolation reagent, glyceraldehyde-3-phosphate dehydrogenase (GAPDH), BCA protein assay kit, bovine serum albumin (BSA, pH 7, $\geq$ 98%), sodium dodecyl sulfate polyacrylamide, skimmed milk, Tris-HCl (pH 7.6), Tris buffered saline (pH 7.4), sodium chloride ($\geq$ 99.9%), monoclonal antibody (p-JNK, p-ERK1/2. JNK, ERK1/2 and NLRP3), Triton-X 100 (0.5%), and 4′,6-diamidino-2-phenylindole (DAPI) were acquired from Sigma Aldrich (Hamburg, Germany). NE-PER^TM nuclear and cytoplasmic extraction reagents, tris-buffered saline (TBS, pH 7.4), enhanced chemiluminescence (ECL) Detection Kit, rat monoclonal antibody NLRP3 and rabbit monoclonal antibody ASC, and horseradish peroxidase-conjugated goat anti-rat were purchased from Thermo Fisher Scientific (Massachusetts, United States). Pure sunflower oil was purchased from local markets in Riyadh, Saudi Arabia.

## Bacterial strains and cell line

Two bacterial strains, *Bacillus subtilis* (*B. subtilis*, ATCC 6633, Gram-positive) and *Pseudomonas aeruginosa* (*P. aeruginosa*, ATCC 27853, Gram-negative) were obtained from the Microbiology Department, King Saud University, Saudi Arabia. The mouse macrophage-derived RAW264.7 cell lines were procured from Microbiology Department, King Khalid Hospital, Riyadh, Saudi Arabia.

## Sample material

White shrimp (*L. vannamei*) shells were purchased from a local market in Riyadh, Saudi Arabia. The shrimps were selected on the basis of maturity and fully grown shells. The shells were thoroughly cleaned with tap water to eliminate the impurities (debris) and hot air-dried at 90°C for 6 h. The shells were blended into homogenized small-sized (<20 mesh) pieces. Samples were kept in refrigerator at- 20°C until further use. Fly ash was obtained from precipitator electrostatic unit from thermal power plant, Riyadh, Saudi Arabia. White transparent polybags were used for packing of the collected materials.

## Isolation of chitin and formation of chitosan

There are few methods reported in the literature for the deproteinization of chitin and formation of chitosan which includes, ultrasound deproteinization of chitin [53], enzymatic deproteinization and demineralization using natural deep eutectic solvents for production of insect chitin from shrimp head waste [54] and most common procedures is enzymatic deproteinization [55]. But we applied more simple chemical demineralization and deproteinization process [56]. Briefly, the dried homogenized shrimp shells (100 g) were demineralized (removing calcium carbonate and phosphate) by adding 1.0 mol $L^{-1}$ hydrochloric acid (1L) at room temperature under agitation for 12 h at 250 rpm. The demineralized shrimp shells were filtered using filter paper followed by several rounds of washing with distilled water to neutralize the pH. Samples were bleached by immersing in 95% ethanol for 10 min to obtain pure crystalline form, and then dried at 75°C in an oven. The dried demineralized shells were subjected to deproteinization by adding 1.0 mol $L^{-1}$ sodium hydroxide (NaOH solid/Liquid, 1:10, g $mL^{-1}$). At 80°C, the mixture was heated for 3 h, the resulting solid was filtered, and washed multiple times with distilled water until neutral pH was attained. After further bleaching, the obtained samples were immersed in ethanol for 10 min to get the chitin (Scheme 1a). Finally, the resulting chitin was oven-dried at 75°C for 8 h. The isolated chitin was treated with 12.5 mol $L^{-1}$ sodium hydroxide (NaOH solid/liquid, 1:15, g $mL^{-1}$) for deacetylation [56]. The mixture was cooled and placed in the freezer for 24 h. Afterwards, the mixture was again heated at 120°C and centrifuged for 5 h at 250 rpm to obtain chitosan (CS). The formed chitosan (CS) was filtered, neutralized with distilled water, and oven-dried at 75°C (Schemes 1b).

(a) Schematic illustration for chitin isolation and production of chitosan and (b) chemical structure of chitosan after deacetylation of chitin.

The degree of CS deacetylation was measured by titration method. Briefly, 0.125 g of CS was dissolved in 0.1 mol $L^{-1}$ hydrochloric acid under constant stirring for 30 min. The prepared CS solution was titrated against 1.0 mol $L^{-1}$ sodium hydroxide solution. A titration curve of NaOH volume vs. pH was generated. The deacetylation degree was determined following the equation:

$$NH_2 = \frac{(C_1V_1 - C_2V_2) - 0 \cdot 016}{G \cdot (100 - W)} \times 100 \qquad (1)$$

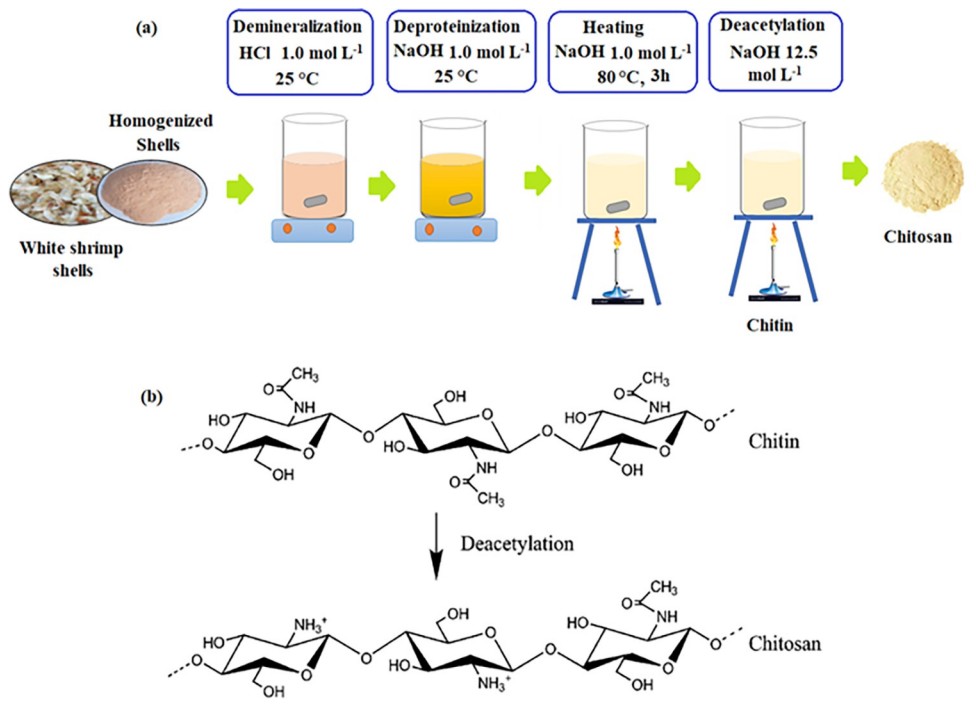

**Scheme 1.**

Where $C_1$, $C_2$, $V_1$, $V_2$, G, W represent the hydrochloric acid concentration (mol $L^{-1}$), the sodium hydroxide concentration (mol $L^{-1}$), the volume of hydrochloric acid ($L^{-1}$), the volume of hydroxide sodium ($L^{-1}$), the sample weight (g), and the water % of the sample, respectively.

The degree of deacetylation was then calculated by applying the following equation:

$$\text{Deacetylation degree}\% = \frac{NH2\%}{9.94\%} \times 100 \qquad (2)$$

## Preparation of fly ash-chitosan nanoparticles (FA-CSNPs)

The ionic gelation procedure was applied for the synthesis of FA-CSNPs by following the protocol of 77 with minor modification [57]. Briefly, chitosan powder (10 g) was dissolved in glacial acetic acid (1.0%, w/v) under constant stirring at 500 rpm for 3h. Then, 200 µL of ash extract solution (5 g dissolved in 50 mL distilled water) was added to the reaction mixture until a grayish solution was obtained. The FA-CSNPs were formed by dropwise addition of 5.0 mL of sodium tripolyphosphate (NTP) solution as a crosslinker agent to 10 mL of chitosan solution under constant stirring for another 30 min. The obtained FA-CSNPs suspension was sonicated for 5 min and freshly formed FA-CSNPs were filtered using an Ultra centrifugal filter and stored at 4˚C in refrigerator prior to further use.

## Synthesis of sunflower oil/fly ash-chitosan bionanocomposite film

The fabrication of ultrafine polymeric sunflower oil-CS plain film was conducted by dissolving 20% (w/v) of CS in 1% (v/v) of glacial acetic acid for 24 h under constant magnetic agitation. Afterwards, approximately 2% (w/v) of sunflower oil of emulsifier teen-80 was added to the CS solution at ambient temperature for 12 h with continuous magnetic agitation at 250 rpm. The obtained uniform film was then treated with 20% (w/v) of FA-CSNPs solution (20 g FA-CSNPs dissolved in 100 mL acetone) for 6 h at ambient temperature under vigorous

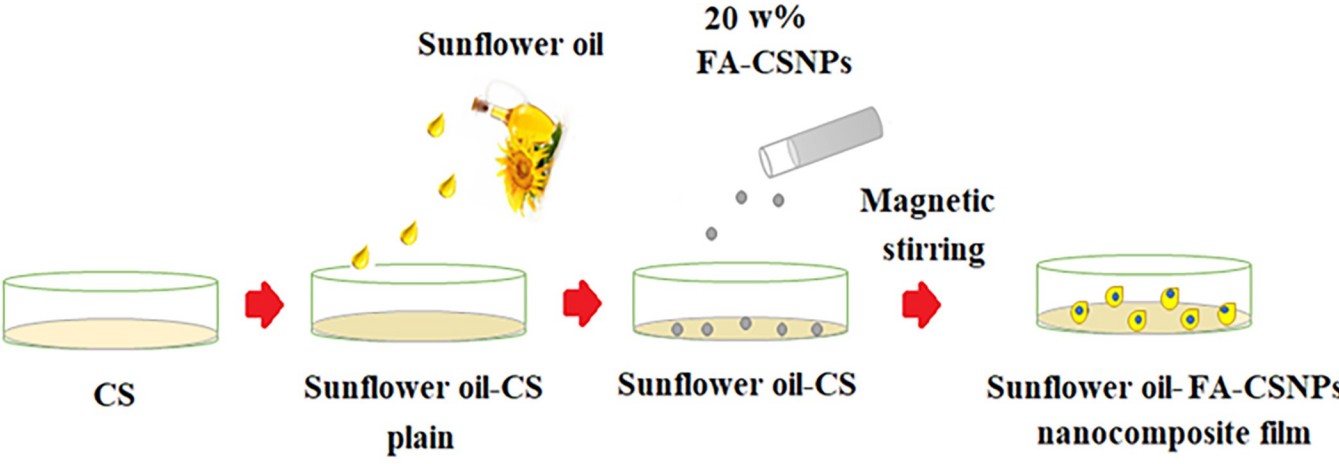

**Scheme 2. Synthesis of sunflower oil–FA–CSNPs nanocomposite film.**

magnetic agitation to get a homogenous polymeric sunflower oil/FA-CSNPs bionanocomposite film. The resulting bionanocomposite film was dried in a desiccator under shade for further experiments (Scheme 2).

## Characterization of FA-CSNPs and sunflower oil/fly ash-chitosan bionanocomposite film

The formation of FA-CSNPs and sunflower oil/FA-CSNPs bionanocomposite film was confirmed by different spectroscopic and microscopic analyses. Fourier-transform infrared (FTIR) spectroscopy (Perkin Elmer Liantrisant, United Kingdom) and X-ray diffraction (XRD, X-ray diffractometer, D/MAX-2500, Tokyo, Japan) were carried out to investigate the possible functional groups and the crystalline surface of the formed FA-CSNPs and bionanocomposite film, respectively. The XRD pattern was performed to verify the existence of FA-CSNPs in the film using Bragg angles ranging from 20˚-70˚, 45 mA, and 30 kV voltage. The morphological shape and size of nanoparticles as well as the bionanocomposite film were investigated by scanning electron microscope (SEM, JSM-7610F, JEOL, California, United States).

## Thermogravimetric analysis (TGA) of bionanocomposite film

Thermogravimetric analysis was performed to measure the thermal stability of the sunflower oil/FA-CSNPs bionanocomposite film using a TGA-502-analyzer (Shimadzu, Tokyo, Japan). Approximately 5 mg of the investigated film was heated from 30 to 500˚C under steady nitrogen gas flow (50 cm$^3$/min) at a heating rate of 25˚C/min to acquire thermogravimetric estimation free from oxidative decomposition. The TGA differential values were used to calculate TGA derivative (DTGA), measured using a forward finite difference protocol.

$$DTGA = (w_{t+\Delta t} - w_t)/\Delta_t$$

Where $w_{t+\Delta t}$ and $w_t$ expressed the residual weight of the test samples at $t + \Delta_t$ time and t, respectively. The $\Delta_t$ represents the time interval for the residual test sample weight reading.

## Antibacterial activity

The antibacterial effect of FA, FA-CSNPs, and sunflower oil/FA-CSNPs bionanocomposite film was examined by the disk diffusion technique [58] against the *B. subtilis* and *P. aeruginosa*.

Briefly, 18 h Mueller-Hinton agar culture plates were used and the microorganism suspension ($1.0\times10^8$ UFC/mL) was adjusted using a sterilized saline solution at 37˚C. Sterilized swabs were dipped in bacterial suspensions and loaded to obtain homogenous bacterial growth on both test and control plates. Approximately, 10 μL of each test sample was added to sterile disks individually. Each sample was loaded on the surface of inoculated plates with 20 μL of microorganisms and subjected to 15 min incubation at room temperature, followed by further incubation for 24 h at 37˚C. The zones of inhibition were recorded in millimeters (mm). A ciprofloxacin-containing disk (5 μg/ disk) was applied as the standard control. Experiments were conducted in triplicates (n = 3). The mean diameter of inhibition was measured and antibacterial effect was categorized as either resistant or sensitive with 10 mm as the cutoff value.

## Estimation of minimum inhibitory and bactericidal concentration

The antibacterial potential with inhibition zone diameter of FA, FA-CSNPs, and bionanocomposite film was tested against each bacterial strain. The broth microdilution procedure using Mueller-Hinton broth in a 96-microwell plate was applied to determine the minimum inhibitory concentration (MIC). The solubility of the tested samples in the broth was increased by adding Tween 80. The test samples (5–1280 μg mL$^{-1}$ range) were applied [59]. Each plate was inoculated with 10 μL of the bacterium inoculum and serial dilution of FA, FA-CSNPs, and the bionanocomposite film. The Mueller-Hinton broth, FA, FA-CSNPs, and bionanocomposite film were applied as negative controls. Meanwhile, the applied positive control was Mueller-Hinton broth with a bacterial inoculum. MICs were measured after 24 h of incubation at 37˚C. The results were determined by using an ELISA reader (Bio Tek, Jinan, Shandong Province, China) at 600 nm. The minimum bactericidal concentration (MBC) of test sample that can lead to complete death of bacteria, was measured using a sterile L rod to plate amounts of first turbid and no growth tubes. The resulting samples were consistently dispersed on nutritional agar plates and kept at 37˚C for 12 h.

## Morphological changes of *B. subtilis* and *P. aeruginosa* (SEM)

The impact of FA, FA-CSNPs, and bionanocomposite film on the shape of both treated and untreated *B. subtilis* and *P. aeruginosa* were examined by scanning electron microscope (SEM, JSM-7610F, JEOL, California, United States). The treated pathogens were cut into 10 mm pieces and fixed in glutaraldehyde solution (3% in phosphate-buffered saline, pH 7.4) for 1 h, then fixed for another 1h in 2% aqueous osmium tetroxide. The collected tissues were dehydrated using graded series (30%, 50%, and 70%) of 95% ethanol and dried with the supply of $CO_2$. The dried tissues were placed on aluminum specimen mount stubs with the help of a silver paint vacuum and observed under SEM at an accelerating voltage of 15 kV.

## Immunomodulatory activity

**Cell viability assay.** Cell viability is an estimation of the amounts of live cells within a population. The determination of cell viability plays a vital role in all types of cell culture. It can be utilized to correlate cell behavior to cell number, giving a more accurate idea of cell metabolism [60]. The effect of FA, FA-CSNPs, and sunflower oil/FA-CSNPs bionanocomposite film on the viability of macrophage RAW264.7 cell line was measured by the MTT assay. Dulbecco's modified eagle medium (DMEM) provided with heat-inactivated 10% of fetal bovine serum (FBS) supplemented with streptomycin (100 μg mL$^{-1}$) and penicillin (100 U mL$^{-1}$) was used to culture the RAW264.7 cells under constant supply of $CO_2$ in a humidified incubator at 37 ˚C. Cells with an exponential growth phase were applied for this study. Briefly, cells ($1.0\times10^5$ cells/mL) were seeded for 24 h into 96-well plates, followed by adding various

concentrations (50, 200, 500 μg mL$^{-1}$) of the tested samples and incubated for another 24 h. The MTT reagent (20 μL) was then poured into each well to make a concentration of 500 μg mL$^{-1}$ and kept again for 4 h incubation. After incubation, the generated formazan crystals were solubilized in DMSO, and a microplate ELISA reader was employed to record absorbance at 570 nm.

## Pinocytic effect

The pinocytic effect of FA, FA-CSNPs, and sunflower oil/FA-CSNPs bionanocomposite film on RAW264.7 cells was determined by a modified neutral red assay method [61]. The RAW264.7 cells at $1\times10^4$ cells/well density were seeded into 96-well plate and incubated in a humidified condition for 24 h (37˚C, 5% CO$_2$). Then, the DMEM medium, lipopolysaccharide (LPS), and varying concentrations (50, 200, and 500 μg mL$^{-1}$) of the test samples were added individually into each well, followed by another 24 h incubation at 37 ˚C. Each concentration was repeated 6 times. Then, the culture medium was replaced, followed by the addition of 100 μL/well of neutral red (0.09%) and further incubated for 30 min. After 30 min incubation, phosphate-buffered saline (PBS, pH 7.4) was used to wash the cells three times, and a 150 μL cell lying solution (ethanol: glacial acetic acid, 1:1 *v/v*) was poured into the wells. The cells were cultured at ambient temperature for 1 h and ELISA reader was used to record the absorbance at 540 nm, and the rate of phagocyte phagocytosis was measured by using the following equation:

$$\text{Phagocyte phagocytosis rate (\%)} = A_S/A_0 \times 100\%$$

Where $A_s$ and $A_0$, represents the absorbance of the tested sample treated and blank groups, respectively.

## Phagocytic uptake estimation

The macrophage phagocytic effect was investigated using the Vybrant$^{TM}$ Phagocytosis Assay kit (*E. coli*, FITC-labeled), ELISA fluorescence microplate reader, fluorescence microscope (Agilent Technologies, Santa Carla, United States), and flow cytometry by following Wang et al. method with slight modification [62]. In brief, RAW264.7 cells with $1 \times 10^5$ cells/well density was plated in 96-microwell plates and incubated for 24 h under 5% CO$_2$ flow at 37 ˚C. Different concentrations (50, 200, and 500 μg mL$^{-1}$) of the test samples or LPS (1.0 μg mL$^{-1}$) were used to treat the cells and incubated for 24 h. Thereafter, the medium was removed with phosphate-buffered saline (1.0 mL) containing 100 μL of the FITC-labeled *E. coli* (1.0 mg mL$^{-1}$), followed by 30 min incubation at 37 ˚C. The cells were then washed thrice with PBS (pH 7.4). At 490 nm excitation and 520 nm emission, absorbance was recorded by fluorescence microplate reader. Meanwhile, the macrophage RAW 264.7 cells with $2\times10^6$ cell/well density in 6-well plates were pre-incubated for 24 h and then exposed to test samples or LPS and incubated further for 6 h. The medium was then removed with PBS (1.0 mL) containing 100 μL of FITC-labeled *E. coli* (1000 μg mL$^{-1}$) and incubated for 30 min at 37 ˚C. The process of phagocytosis was stopped by adding ice-cold PBS (2.0 mL) and the cells were washed thrice again with cold PBS (pH 7.4). The CellQuest software was applied for the flow cytometric estimation on the FACScan flow cytometer (Beckham, San Jose, California, United States). For the image analysis of the test sample-mediated phagocytosis, the 96-well plates were loaded with RAW 264.7 cells with $1\times10^5$ cell/well density for 24 h. The cells were then stimulated with test samples or LPS for 6 h, followed by the addition of FITC-labeled *E. coli* solution (1000 μg mL$^{-1}$) and incubated at 37 ˚C for 30 min. Finally, the cells were fixed with 4% paraformaldehyde solution and viewed under the fluorescence microscope.

## Estimation of NO, TNF-α, IL-6 and IL-1β cytokine production

24-well plates were loaded with $4 \times 10^5$ cells/well density of RAW264.7 cells cultured with different concentrations (50, 200, 500 μg mL$^{-1}$) of test samples for 24 h. The supernatants were then collected. The accumulation of nitrite in the supernatant was determined as an indicator of NO generation as indicated by the Griess reaction. The concentration of cytokines (TNF-α, IL-6, and IL-1β) was measured by applying the commercial enzyme-linked immunosorbent ELISA kits assay following the instructions of the manufacturer. The cytokine concentration was calculated using standard curves.

## Real time-polymerase chain reaction (RT-PCR) analysis

TRiPure$^{MT}$ isolation reagent was used to extract the total RNA according to the manufacturer's instructions. Briefly, 3 g of total RNA was applied for the synthesis of complementary DNA using reverse transcriptase reactions. iNOS, IL-6, and TNF-α genes encoded reverse transcriptase-generated complementary DNA were enhanced using specific primers. Invitrogen was used to design and synthesize the primers with the following nucleotide sequences: (Forward: 5´–CGGCAAACATGACTTCAGGC–3´; Reverse: 5´–GCACATCAAAGCGGCCATA G–3´), IL-6 (Forward: 5´–TACTCGGCAAACCTAGTGCG–3´; Reverse: 5´–GTGTC CCAA CATTCATATTGTCAGT–3´), TNF-α (Forward: 5´–GGGGATTATGGCTCAG GGTC–3´; Reverse: 5´–CGAGGCTCCAGTGAATTCGG–3´), GAPDH (Forward: 5´–TTTGTCAAGCT CATTTCCTGGTATG–3´; Reverse: 5´–TGGGATAGGGCCTCT CTTGC–3´). All samples under investigation were examined three times. The internal control employed was Glyceraldehyde-3-phosphate dehydrogenase (GAPDH). The expression level of mRNA for these cytokines was estimated by $2^{-ΔΔ}$ CT protocol [63].

## Western blot analysis

A standard procedure was carried out for the western blot analysis [64]. 6-well plates were seeded with RAW264.7 cells at $1.0 \times 10^6$ cells/well density and incubated for 24 h at ambient temperature. The cells were then treated for 12 h with LPS or varying concentrations (50, 200, and 500 μg mL$^{-1}$) of test samples. The RAW264.7 cells were washed with cold PBS (pH 7.4) twice, followed by NE-PER$^{TM}$ nuclear, and cytoplasmic extraction reagents. BCA protein assay kit was applied to measure the protein contents using bovine serum albumin (BSA) as reference. Gel electrophoresis was applied to separate the denatured proteins using 10% of sodium dodecyl sulfate-polyacrylamide and transferred to a polyvinylidene difluoride (PVDF) membrane filter for Western Blotting. After membrane blocking with 5% skimmed milk containing 20 mM of tris-HCl (pH 7.6), tris-buffered saline (pH, 7.4), 150 mM of NaCl, and 0.1% Tween-20, the membrane was incubated for 30 min at ambient temperature. The obtained blot was further incubated overnight at 4 ˚C with target monoclonal antibody (p-JNK, JNK, p-ERK1/2. ERK1/2, and NLRP3) in TBS containing 5% of BSA. Subsequently, TBS was used to wash the membrane multiple times. The membrane was then incubated for another 1 h with an appropriate secondary antibody horseradish peroxidase-conjugated goat anti-rat. The membrane was washed again three times with TBS for 5 min. Signals were monitored with ECL Detection Kit with bands captured using Molecular Image ChemiDocXRS + Imagine system (BIO-RAD Laboratories, Hercules, United States).

## Confocal analysis

Immunofluorescence staining was performed to detect the colocalization of NLRP3 with apoptosis-associated specks induced responses to inflammation (ASC) using a confocal laser

scanning microscope (ZEISS, LM 880, Jena, Germany). The RAW264.7 cells with $10^5$ density were plated onto a confocal plate and incubated under a humidified condition with a $CO_2$ supply at 37 ˚C for 24 h. After treatment with various concentrations of test samples or LPS for 12 h, cold PBS (pH 7.4) was used to wash the cells three times and glued with cold 4% paraformaldehyde solution for 15 min at ambient temperature. Approximately, 100 μL of Triton-X 100 (0.5%) was poured into each plate at ambient temperature for 10 min. Subsequently, cold PBS was used to wash the cells multiple times for 5 min and then blocked with BSA (5%) for 30 min at room temperature. After blocking, the cells were incubated with 20 μg mL$^{-1}$ of rat monoclonal antibody NLRP3 and rabbit monoclonal antibody ASC (1:200) in PBS containing BSA (5%) at 4 ˚C for 12 h. After another wash with PBS, the cells were then treated with horseradish peroxidase-conjugated goat anti-rat or anti-rabbit IgG (2ry-antibody) and incubated for 1 h. The cells were further washed for 5 min followed by the addition of 50 μL of a fluorescence staining solution DAPI and incubated for 15 min at 25 ˚C. Lastly, the cold PBS was applied to wash the cells three times and the cells were pictured under a confocal laser scanning microscope.

## Statistical analysis

The provided data was expressed as means ± standard deviation (±SD). Analysis of variance was used to establish statistical significance, followed by the student's t-test. A $P$-value $< 0.05$ was considered as the significance level.

## Results and discussion

### Characterization of sunflower oil/fly ash-chitosan bionanocomposite film

The homogeneous sunflower oil/fly ash-chitosan bionanocomposite film was prepared by treating FA-CSNPs solution with sunflower oil under continuous magnetic stirring at 250 rpm for 12 h. The formation polymeric sunflower oil/FA-CSNPs bionanocomposite film as well as FA-CSNPs were identified and characterized using various spectroscopic and microscopic techniques. The FTIR spectra of FA, CS, FA-CSNPs, and Sunflower oil/FA-CSNPs bionanocomposite film were demonstrated in Fig 1. The FTIR spectrum of raw fly ash exhibited broad absorption peaks at 3449 and 1634 cm$^{-1}$ assigned to O-H and H-O-H groups stretching and deformation vibrations of the water molecules [65]. A strong absorption peak at 1385 cm$^{-1}$ noticed in fly ash spectrum was ascribed to C-O deformation bending vibration. While as, the absorption band displayed at 1074 cm$^{-1}$ was attributed to the Si-O asymmetric stretching in tetrahedra. The presence of mullite in the fly ash was indicated by the prominent absorption peak at 567 cm$^{-1}$ (Fig 1A). The main functional moiety of observed in FTIR spectrum of chitosan was indicated by absorption peaks at 3455, 2875 and 2372 cm$^{-1}$ due to O-H group stretching vibration, the medium stretching vibration of C-H and atmosphere O = C = O of $CO_2$, respectively. However, the existence of absorption peaks at 1604 and 1450 cm$^{-1}$ in the chitosan moiety was due to the bending vibration of N-H of protonated amino (-NH$_2$) group and the medium bending vibration of C-H of the alky group, respectively. Two anti-symmetric stretching vibration at 1083 and 892 cm$^{-1}$ in chitosan were recognized due to C-O-C bridges of and designated to the glucopyranose ring in the chitosan matrix (Fig 1B). The FTIR spectrum of formed FA-CSNPs exhibited less sharp and broad absorption peaks at 3451, 2870, 1642, 1156, 1083, and 569 cm$^{-1}$ due to the medium stretching N-H amine, strong stretching N-H amine salt, N-H bending amine, Si-O -Si/Si-O-Al and Al-O/ Si-O bending vibration, respectively (Fig 1C). Whereas, significant modifications were observed in the FTIR spectrum of as-synthesized sunflower oil/FA-CSNPs bionanocomposite film. A notable vibration band at 3452 cm$^{-1}$ in the bionanocomposite film was assigned to the free O-H bond valence, which is associated to the

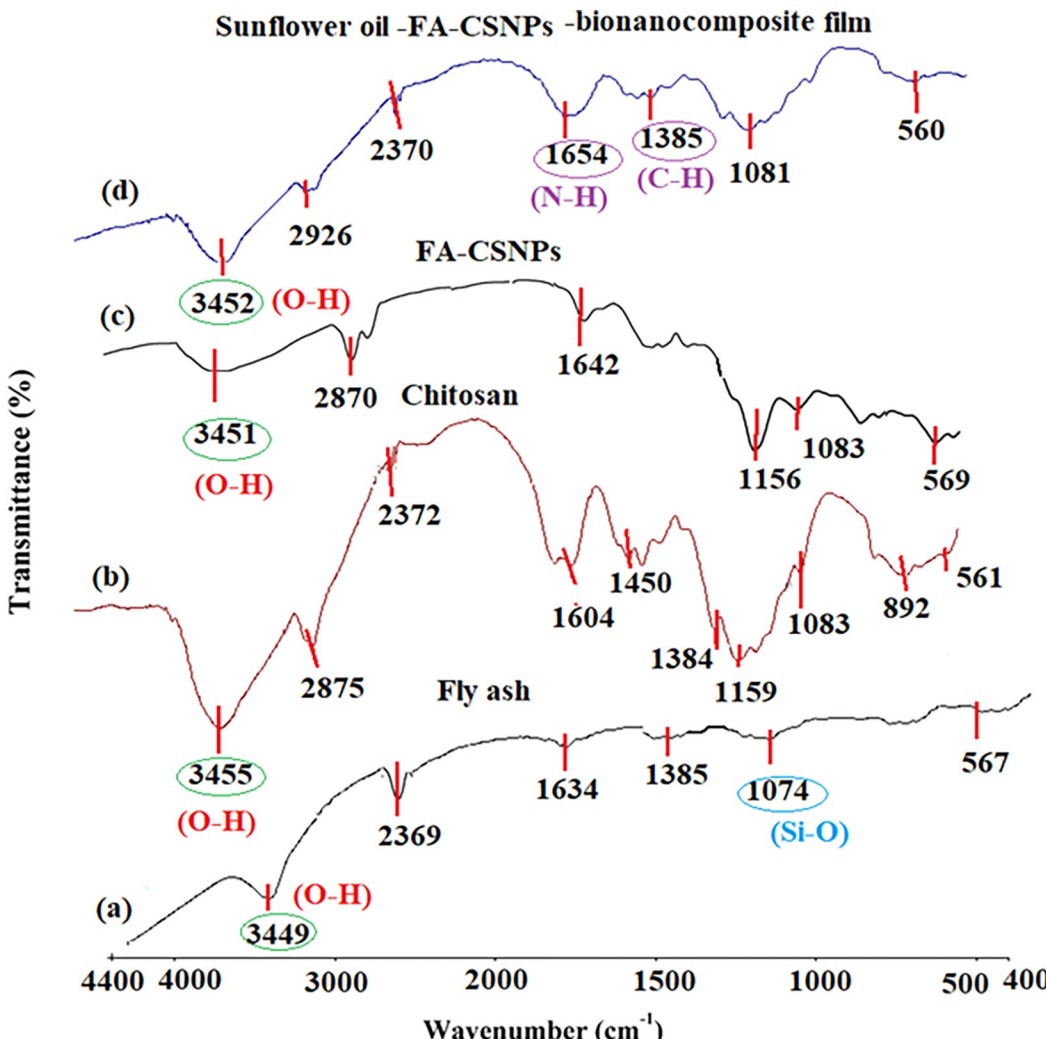

**Fig 1.** FTIR spectra of (a) Fly ash, (b) chitosan, (c) FA–CSNPs, and (d) Sunflower oil/FA–CSNPs–bionanocomposite film measured at wavenumber 4400–400 cm$^{-1}$ using KBr pellets under 10 kg/cm$^2$ pressure for 50 s.

polymeric hydrogen bonds due to existence of broadband in the 3550–3200 cm$^{-1}$ region. The presence of absorption peaks at 1654 and 1385 cm$^{-1}$ in the bionanocomposite film moiety indicates the existence of N-H bending vibration of the protonated amino group (-NH$_2$) and bending vibration of the C-H group of the alkyl. Moreover, most of the vibration bands due to chitosan and fly ash were shifted to higher wavenumbers after the incorporation in sunflower oil. This might be due to the phytoconstituents of oil that can interfere the intra/intermolecular hydrogen bonding found in the neat chitosan and fly ash matrix (Fig 1D).

The crystalline phases of CS, FA, and FA-CSNPs were studied by their XRD patterns. The XRD pattern of fly ash exerted two major crystalline phases including quartz and mullite along with the amorphous constituents on an aluminosilicate glass basis. The fly ash micrograph revealed that the grains of fly ash were irregularly spherical (Fig 2A).

While, the chitosan XRD manograph exhibited two reflection falls at 2θ = 11° and 22° corresponding to crystal I and crystal II of the chitosan structure [66]. Chitosan also displayed strong broad lines for small diffraction angles, suggesting a long disorder range (Fig 2B).

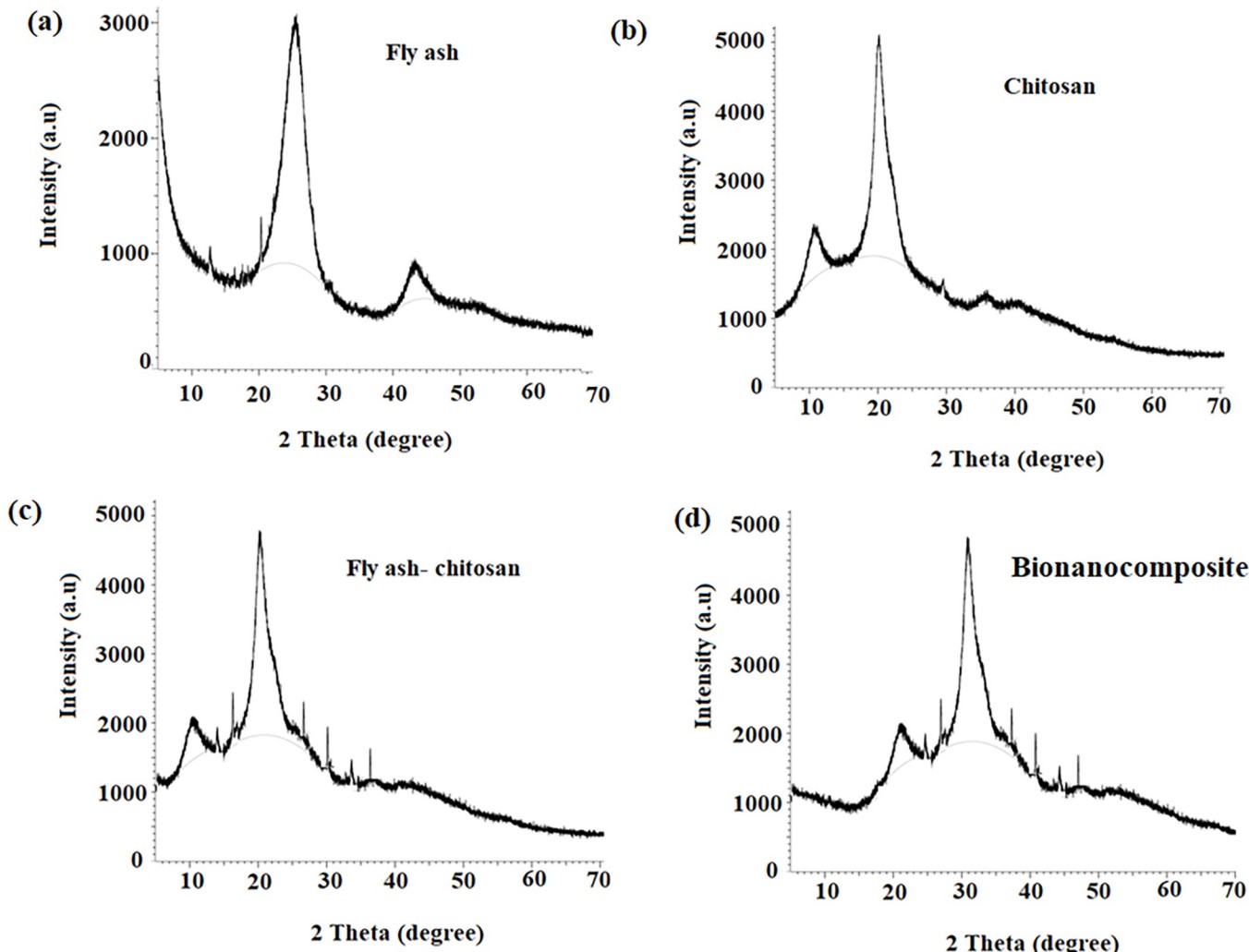

**Fig 2.** XRD pattens of (a) fly ash, (b) chitosan, (c) fly ash–chitosan nanoparticles, and (d) sunflower–oil–fly ash–CSNPs bionanocomposite film measured at 2 theta degree in the range of 10–70˚.

However, the crystallinity of pre-synthesized FA-CSNPs was observed at angles (2θ) at 12.65˚, 15.21˚, 19.04˚, 23.26˚, 34.82˚, and 37.23˚ in the XRD spectrum (Fig 2C). The crystalline size of FA-CSNPs was found to be 6.90 nm when calculated using Scherrer's equation [67]:

$$D = 0.9\lambda/\beta \cos\theta$$

Where λ (x-ray wavelength), θ (Bragg diffraction angle), and β (full width at half maximum of the XRD peak appearing at a diffraction angle θ), respectively. The XRD pattern of sunflower oil/FA-CSNPs-bionanocomposite film exhibited the diffraction peaks at same angles as observed in the XRD spectrum of FA-CSNPs (Fig 2D).

SEM coupled with EDS was used to study the shape and elemental analysis of FA, CS, FA-CSNPs, and sunflower oil/FA-CSNPs-bionanocomposite film (Fig 3). The SEM images of FA surface morphology at different magnifications showed two types of spheres cenospheres and thin walled plerosheres with smaller ones enveloped or minerals inside these spheres (Fig 3A). Whereas, the images of chitosan showed smooth and clear surface (Fig 3B). Whereas, the loading of FA on the surface of CS was clearly observed by the occurrence of cenospheres and

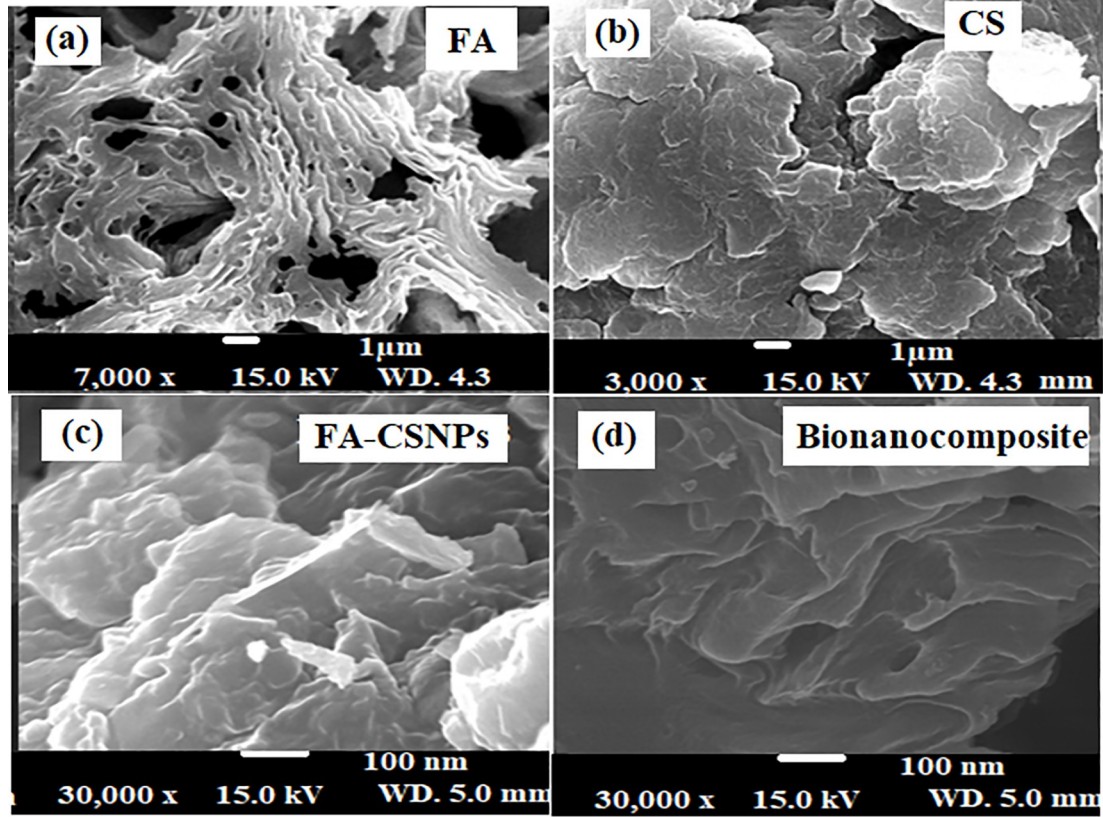

**Fig 3.** SEM images of (a) FA at 7,000 x, (b) CS at 3,000 x, (c) FA–CSNPs at 30,000 x, and (d) sunflower oil–FA–CSNPs bionanocomposite film at 30,000 x magnification.

plerosheres in the picked images of FA-CSNPs (Fig 3C). However, the surface morphology of sunflower oil-FA-CSNPs bionanocomposite film appeared compact and denser as compared to FA-CSNPs due to sunflower oil incorporation. The incorporation of FA-CSNPs in the film showed the structure was homogenous, even and without cracks and pores (Fig 3D).

The elemental analysis includes the EDS and mapping of the elements as demonstrated in Fig 4A–4D. The observed results revealed the weight % of Zn, Cu, O, C in FA sample were 0.5, 5.1, 10.0, and 82.8%, respectively. Whereas, in CS the weight % was found to be 0.6, 1.0, and 39.3, and 59.1% for Zn, Cu, O, and C, respectively. However, the weight % O and C in the FA-CSNPs was found to be 36.3 and 61.6% as well as traces of S (2.2%). While as, the weight percentage of O (45.2%) and C (68.3%) was highest in bionanocomposite film besides containing Zn (0.7%), Cu (6.3%), N (5.7%), and traces of S (2.1%), which might be attributed to the existence of phytoconstituents in the sunflower oil.

## Thermogravimetric analysis of sunflower oil-FA-CSNPs bionanocomposite film

The thermogravimetric analysis (TGA) involves constant monitoring of the mass of the test sample as a function of temperature and time during the passage of the inert-gas environment over the sample. A dynamic TGA was conducted for the bionanocomposite film. The percentage of weight loss was noted between 30°C to 500°C under the $N_2$ gas atmosphere at the rate of 10°C min$^{-1}$ [68]. A multi-step thermal degradation was noticed in the bionanocomposite film

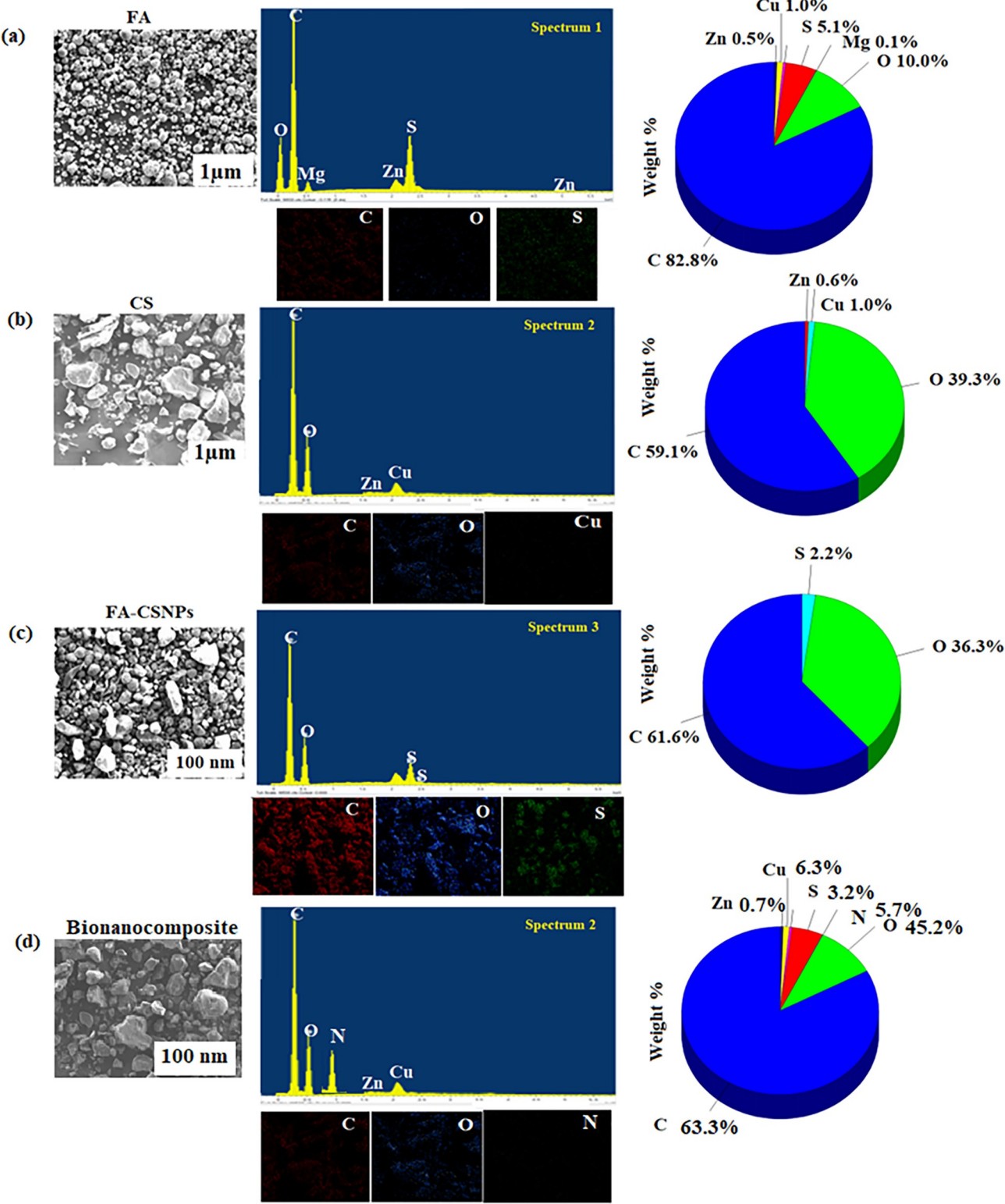

**Fig 4.** SEM and EDS images, elemental mapping, and weight % of (a), FA. (b) CS, (c) FA–CSNPs, and (d) sunflower oil–FA–CSNPs bionanocomposite film.

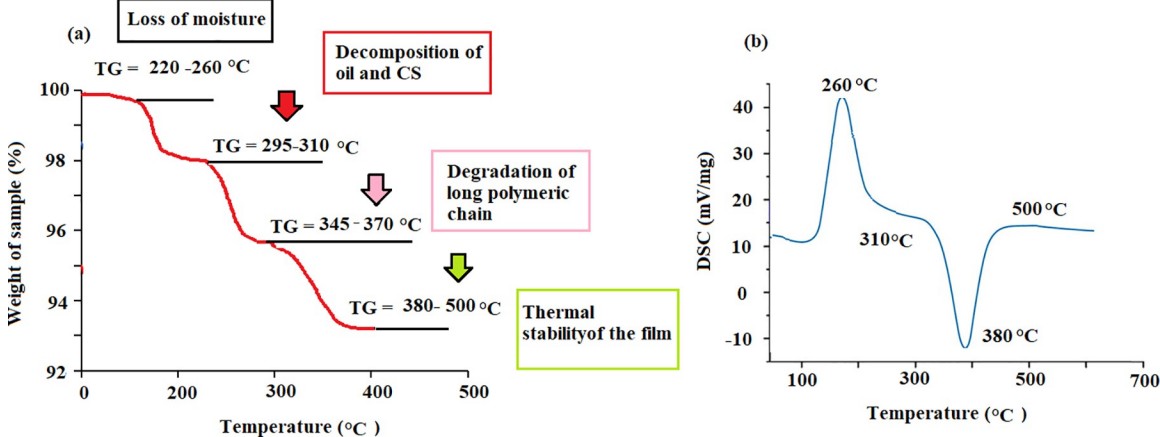

**Fig 5.** (a) TGA and (b) DTGA of sunflower oil/FA–CSNPs bionanocomposite measured in the range from 30 to 500˚C.

due to the thermal degradation behavior of chemical constituents of CS, FA, and sunflower oil. As illustrated in the TGA graphs (Fig 5), the thermal decomposition process of CS, FA, and sunflower oil took place through several steps. The initial degradation of bionanocomposite film started with the evaporation of 10–15% of moisture content and the total weight loss occurred at approximately 100˚C. The thermal degradation continued at 220˚C and 260˚C and exhibited the maximum decomposition rate at 270˚C and 280˚C for the sunflower oil and CS film, respectively, which was ascribed to the volatilization of the constituents of sunflower oil and fly ash. The main decomposition step began at 295˚C and 310˚C with a maximum decomposition rate at 345˚C and 370˚C, mainly due to the degradation of the long biopolymeric chain. Presumably, the CS-FANPs have a dual impact on the bionanocomposite film, i.e., catalytic effect against polymeric matrix and sunflower oil reduced the thermal stability and barrier effect to increase the thermal stability. The results of this study suggested that the sunflower oil/ FA-CSNPs bionanocomposite film showed stability up to 380˚C and was found higher as compared to the individual components due to their strong intermolecular interaction and sample inherent characteristics (Fig 5A and 5B). The thermogravimetric analysis also confirmed the interaction between organic and inorganic surfaces of nanomaterials [69].

## Antibacterial activity

The FA, FA-CSNPs, and sunflower oil/ FA-CS bionanocomposite film expressed varying degrees of sensitivity towards *B. subtilis* and *P. aeruginosa* bacterial strains. Both the bacterial pathogens were found sensitive to the tested samples. The results showed that the bionanocomposite film exerted potent antibacterial activity against *B. subtilis* and *P. aeruginosa*, followed by FA-CSNPs and FA in a dose-dependent manner (Fig 6). Three varying concentrations (25, 50, and 100 µg mL$^{-1}$) of FA, FA-CSNPs as well as the bionanocomposite film were applied against the two bacterial strains. At 100 µg mL$^{-1}$, the bionanocomposite film exerted larger zone of inhibition in bacterial growth against the two strains when compared with FA-CSNPs and FA. The outcomes showed inhibition zones (10, 11, and 14 mm), (13, 14, and 17 mm), and (25, 26, and 28 mm) for FA, FA-CSNPs, and bionanocomposite against *B. subtilis*, respectively (Fig 6Aa–6Ac and 6Ba–6Bc). However, larger inhibition zones were observed against *P. aeruginosa* (14, 16, and 18 mm), (15, 16, and 18 mm), and (30, 32, and 34 mm) for the three concentrations, respectively (Table 1). The bionanocomposite film exerted strongest antibacterial effect against *P. aeruginosa*. (Fig 6B). The higher antibacterial potential

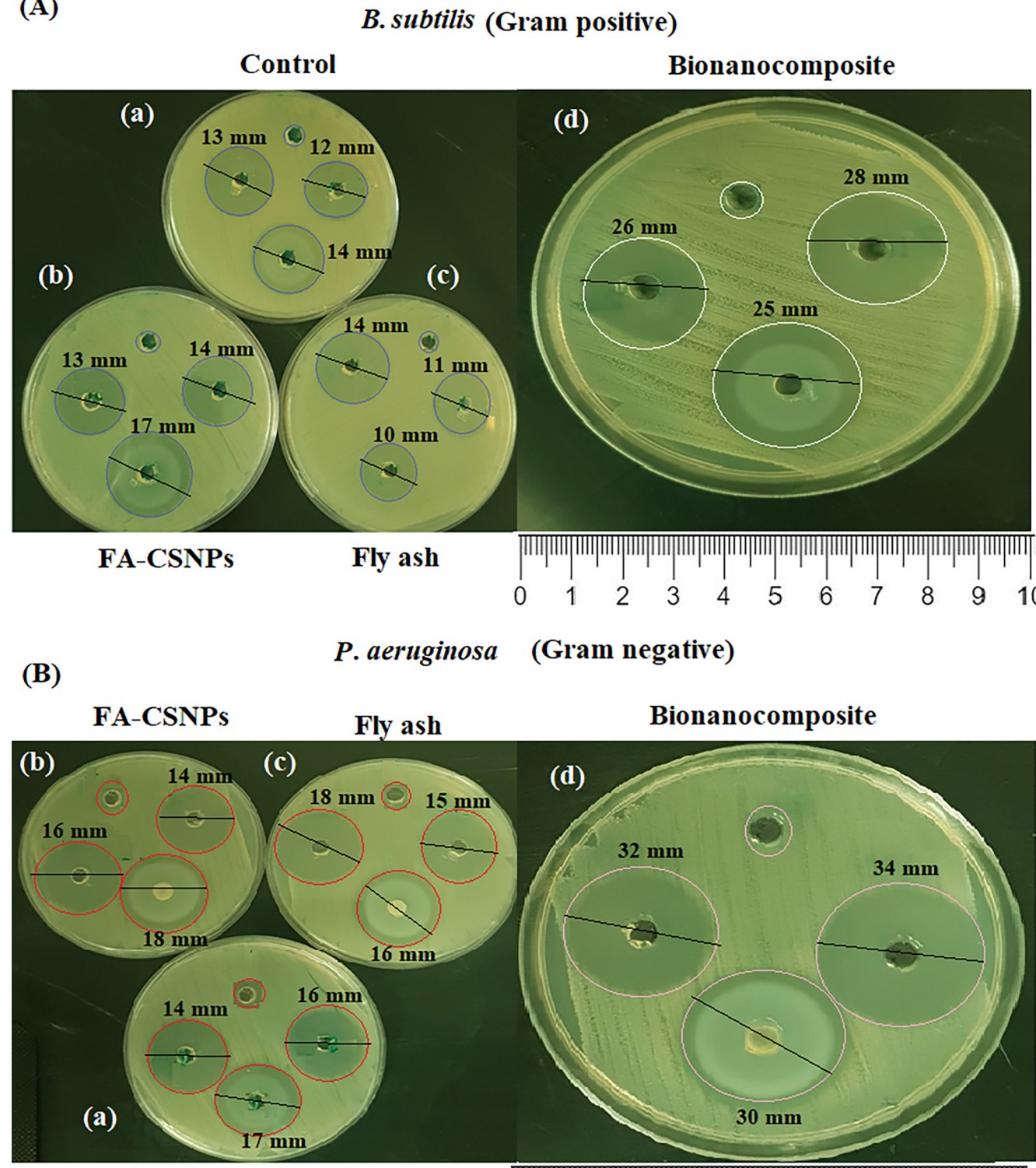

**Fig 6.** Antibacterial potential of (a) control, (b) FA–CSNPs, (c) fly ash, and (d) sunflower oil/FA–CSNPs bionanocomposite film against (A) *B. subtilis* and (B) *P. aeruginosa* strains.

**Table 1.** Antibacterial potential of FA, FA–CSNPs, sunflower oil/FA–CSNPs bionanocomposite against *B. subtilis* (Gram–positive) and *P. aeruginosa* (Gram–negative) bacteria.

| No. | Samples | Antibacterial Potential | | | |
|---|---|---|---|---|---|
| | | *B. subtilis* | | *P. aeruginosa* | |
| | | Concentration (µg mL$^{-1}$) | Zone of Inhibition (mm) | Concentration (µg mL$^{-1}$) | Zone of Inhibition (mm) |
| 1 | Fly ash | 25 | 10 | 25 | 14 |
| | | 50 | 11 | 50 | 16 |
| | | 100 | 14 | 100 | 18 |
| 2 | FA-CSNPs | 25 | 13 | 25 | 15 |
| | | 50 | 14 | 50 | 16 |
| | | 100 | 17 | 100 | 18 |
| 3 | Bionanocomposite | 25 | 25 | 25 | 30 |
| | | 50 | 26 | 50 | 32 |
| | | 100 | 28 | 100 | 34 |
| 4 | Control (Ciprofloxacin) | 25 | 12 | 25 | 14 |
| | | 50 | 13 | 50 | 16 |
| | | 100 | 17 | 100 | 17 |

of FA-CSNPs and bionanocomposite film were dependent on the shape, size, surface area, and morphology of the particles as well as the polarity of the surface.

## Bacteriostatic and bactericidal determination

The sunflower/oil-FA-CSNPs bionanocomposite film exhibited strongest antibacterial potential was subjected to measure bacteriostatic (MIC) and bactericidal (MBC) effects using the agar diffusion method [70] against two bacterial strains *i.e.*, *B. subtilis* and *P. aeruginosa*. MIC is considered the minimum concentration of bionanocomposite film that inhibits the inoculum growth after incubation at 37 °C for 24 h. The gradual increase in concentration ranging from 5 to 1280 µg mL$^{-1}$ of the bionanocomposite film caused a significant inhibition in the cell viability of bacteria ($p < 0.05$). The concentration of 350 µg mL$^{-1}$ was the MIC of the bionanocomposite film against *B. subtilis* and *P. aeruginosa* (Fig 7A and 7B). However, the lowest concentration of bionanocomposite film that led to the complete death of the pathogens under particular conditions throughout a fixed time was known as MBC [71]. The obtained results

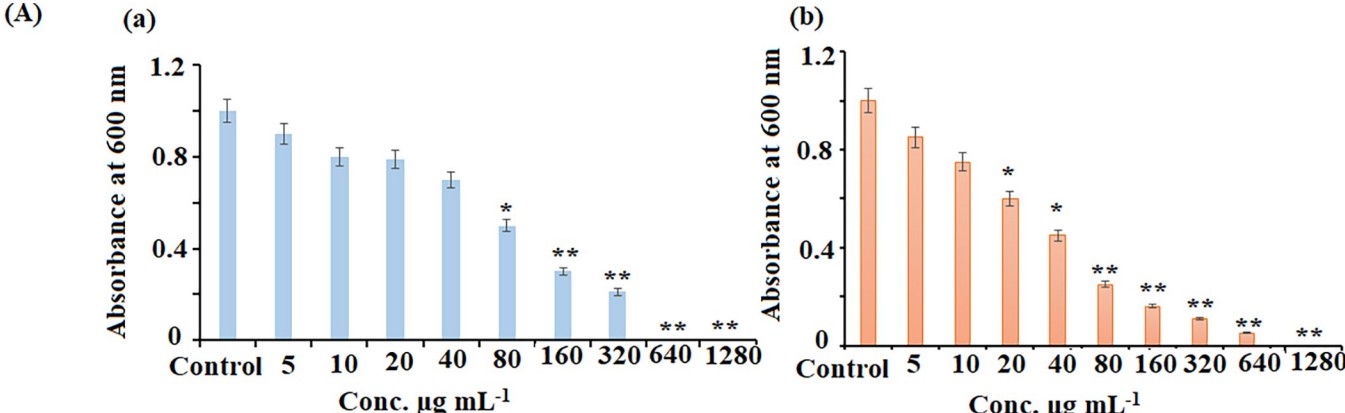

**Fig 7.** Minimum inhibitory concentrations (MIC) of sunflower oil/FA–CSNPs bionanocomposite (µg mL$^{-1}$) against (a) *B. subtilis* and (b) *P. aeruginosa*. The data represented as mean ± standard deviation of thrice measurements.

**Table 2. Minimum bactericidal concentration (MBC) of sunflower oil/FA–CSNPs bionanocomposite against *B. subtilis* and *P. aeruginosa*.**

| Sample | CFU mL$^{-1}$ | |
|---|---|---|
| | *B. subtilis* | *P. aeruginosa* |
| Control | TNTC | TNTC |
| 5 | TNTC | TNTC |
| 10 | TNTC | TNTC |
| 20 | TNTC | TNTC |
| 40 | TNTC | TNTC |
| 80 | TNTC | TNTC |
| 160 | $5\times10^2$ | $5\times10^4$ |
| 320 | 143 | $3\times10^2$ |
| 640 | 8 | NIL |
| 1280 | 1 | NIL |

TNTC: Too numerous to count

revealed that the MBC of the bionanocomposite film for the two investigated bacterial strains were 640 and 1280 μg mL$^{-1}$ for *B. subtilis* and *P. aeruginosa*, respectively (Table 2). The noticed inhibition zones could be due to the penetration of the content of bionanocomposite film inside the bacterial cells causing damage to mitochondria, leakage of cell contents, and subsequently cell death. The possible mechanism of the bactericidal effect of bionanocomposite film including damage to the cell membrane and cell death can be referred to the oxidative stress occurred due to the release of reactive oxygen and nitrogen species (ROS and RNS) or the interaction between the FA-CSNPs ions with the outer lipopolysaccharide (LPS) layer of the bacterial cell membrane [72]. Furthermore, the antibacterial effect of FA, FA-CSNPs and bionanocomposite film against *B. subtilis* and *P. aeruginosa* demonstrated that all the investigated samples were more susceptible to *P. aeruginosa* (Gram-negative) than *B. subtilis* (Gram-positive) bacteria. This could be attributed to the differences in content and thickness of their cell wall membranes. Gram-positive strains have a cell wall thickness ranging from 20 to 80 nm compared to less than 10 nm in the Gram-negative ones [73]. As a result, the cell walls of *P. aeruginosa* contained a thin peptidoglycan layer, and consequently the antibacterial activity of the bionanocomposite film was excellent.

## Morphological study of *B. subtilis* and *P. aeruginosa* using SEM

SEM was used to study the influence of FA, FA-CSNPs and the bionanocomposite film on the surface morphology of *B. subtilis* and *P. aeruginosa*. Amongst the three, the shape and size of bacterial cell surfaces were significantly damaged after application of bionanocomposite film, followed by FA-CSNPs and FA. The bionanocomposite film showed efficient penetration of the peptidoglycan membrane of bacterial cells, causing membrane breakdown, the release of cell components, and cell death [74]. The efficient penetration of bionanocomposite film could be attributed to the presence of FA-CSNPs and the phytochemical components of sunflower oil on the surface of film that has attacked the pathogen cell membrane and the enhanced greatly the antibacterial action of the film, when compared with FA-CSNPs and FA [75] (Fig 8).

The exact mechanisms of nanoparticles towards pathogen still remain unknown. However, researchers have proposed that the action of nanoparticles on pathogens may due to its ability to enter into the cell, the production of free radicals, the formation of reactive oxygen species and the inactivation of cellular proteins by nanoparticles [76]. Besides that, there are also some

**Fig 8.** The morphological changes of untreated bacterial cells *B. subtilis* and *P. aeruginosa* (a,e) control and treated with (b,f) chitosan, (c,g) fly ash, and (d,h) sunflower oil/FA–CSNPs bionanocomposite under SEM.

other factors affecting the bactericidal mechanisms of nanoparticles including, concentration of nanoparticles and type of bacterial strain, shape, size and combination of different antibiotics [77]. Two possible mechanisms were hypothesized for the antibacterial activity of the produced bionanocomposite film, which could be the photogeneration of reactive oxygen species (hydroxyl ion, superoxide ion, hydrogen peroxide, and singlet oxygen) or the development of electrostatic interaction between the pathogen cell membrane and the generated nanoparticles ions (+ve charge of chitosan interacts with–ve charge of the cell membrane). The nanoparticles ions embedded in the film reacted with the thiol (-SH) or sulfhydryl group on the cell surface and proteins of the cell membrane to form the stable S-metal group, resulting in the detachment of hydrogen ions from the protein molecules, which lowers the tissue permeability and causing the cell death [78]. The second possible mechanism could involve direct attack to the cell membrane by nanoparticles ions and enhancing the formation of ROS species, causing protein, DNA, and lipid damage. In addition to this, one of the most critical mechanisms involved is suppression of enzyme production which caused the disruption of assimilatory food pathway and finally leading to cell murder [79]. The antibacterial action of the bionanocomposite film could be due to the invasion of FA-CSNPs nanoparticles into the bacterial cell causing death (Scheme 3).

## Immunomodulatory activity

The immunomodulatory potential of FA, CS, FA-CSNPs, and the sunflower oil/ FA-CS bionanocomposite film was examined by different parameters.

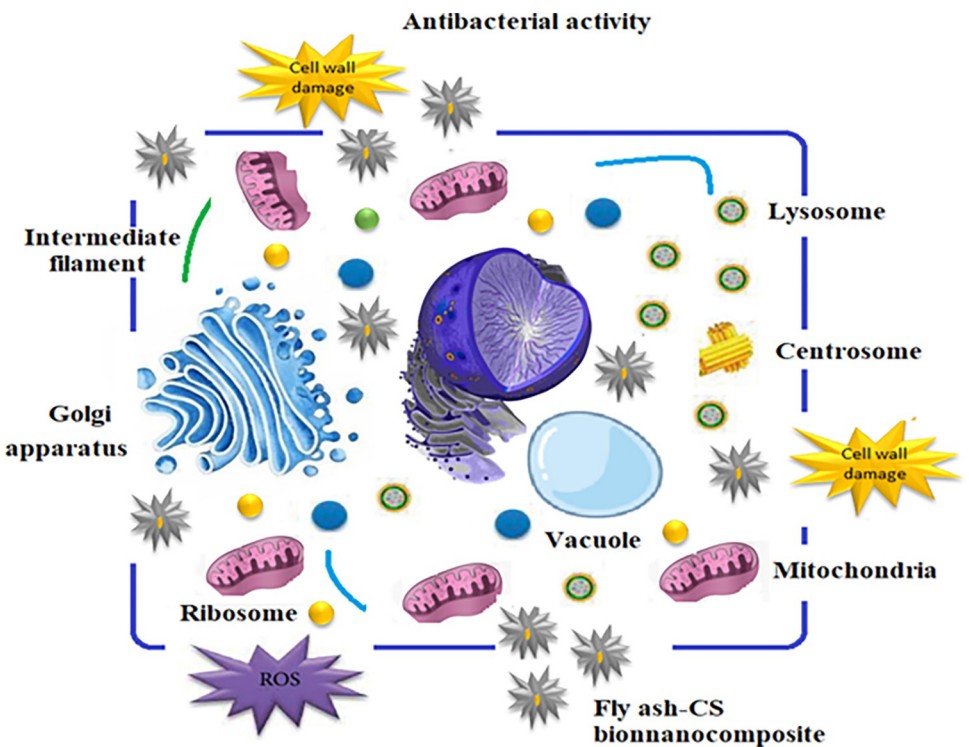

**Scheme 3. The possible mechanism of antibacterial potential of sunflower oil/FA–CSNPs bionanocomposite against the bacterial cells.**

## Effect on RAW264.7 cell viability

Prior to examining the immunomodulatory potential of FA, CS, FA-CSNPs, and bionanocomposite film, their toxic effect on the macrophage of RAW264.7 cells was evaluated by MTT assay. After the treatment with test samples at varying concentrations (50, 200, and 500 μg mL$^{-1}$) for 24 h, significant cell viability was observed (Fig 9A). The results revealed that the highest cell viability effect was exhibited by the bionanocomposite film (98.95%), followed by FA-CSNPs (83.25%) at 200 μg mL$^{-1}$ concentration ($P > 0.05$). Meanwhile, chitosan and fly ash displayed moderate (62.47%) and low (45.24%) cell viability at the same concentration, respectively.

## Pinocytic and phagocytic effect on RAW264.7

The up-regulation of the innate immune response is referred to as macrophage activation and is considered one of the most essential processes in the immune response [80]. One of the most noticeable characteristics of macrophage activation is the rise in phagocytosis such as pinocytic and phagocytic effects [81]. The effect of test samples on the pinocytic and phago-cytic activity was investigated at various concentrations (50, 200, and 500 μg mL$^{-1}$). The results showed that the rate of phagocyte phagocytosis of neutral red in the control group or test sam-ple-induced RAW264.7 cells was 100%, 124.85%, 145.84% and 173.35% for FA, CS, FA-CSNPs, and bionanocomposite film, respectively at 200 μg mL$^{-1}$. Furthermore, the results of the phagocytosis assay kit revealed that the utilization of fluorescence intensity of FITC-labeled *E. coli* was enhanced significantly in bionanocomposite film-treated or LPS-treated and FA-CSNPs-treated or LPS-treated for 24 h compared to untreated RAW264.7 cells (Fig 9B). Also, the green fluorescence was noticed in RAW264.7 cells under a fluorescence

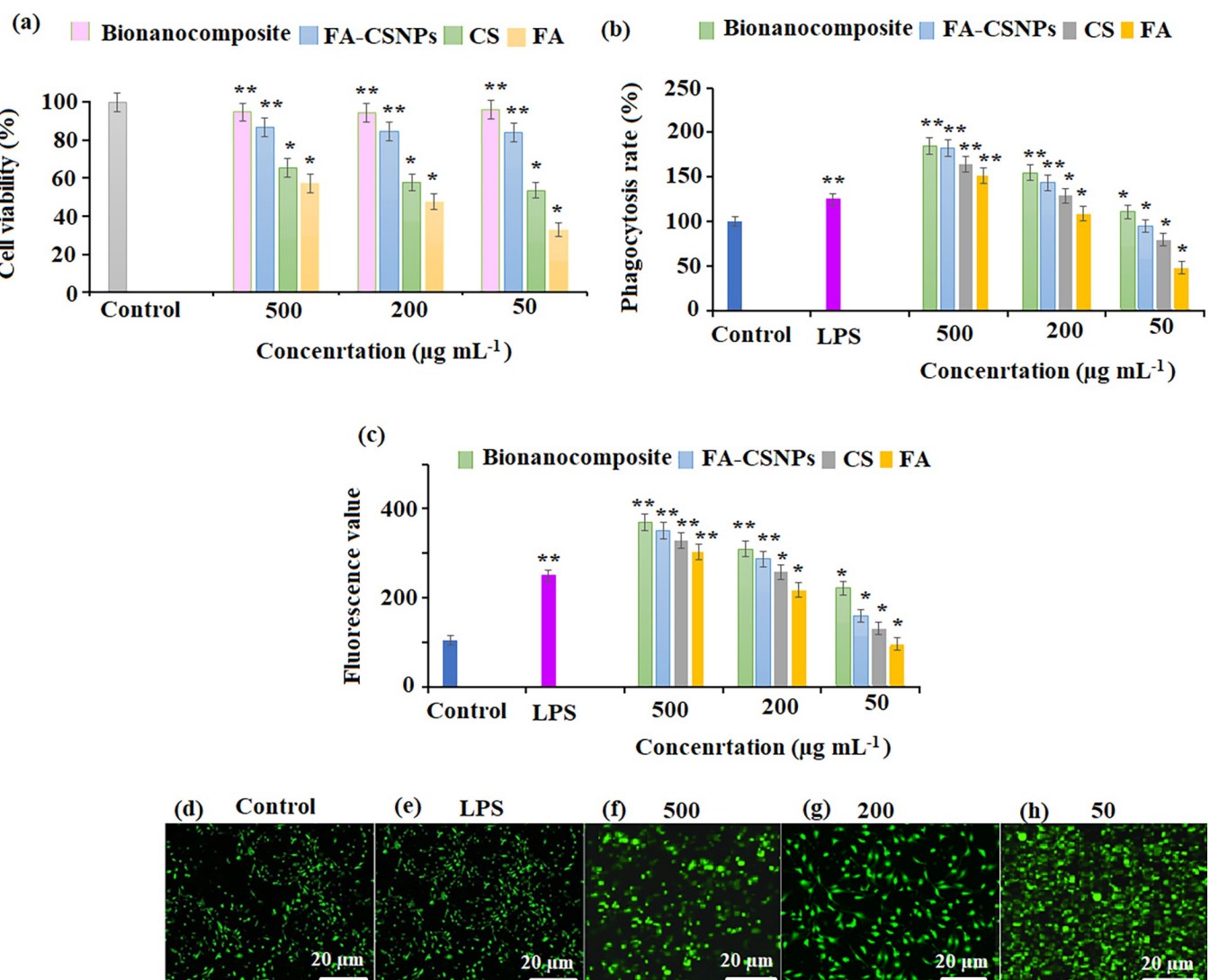

**Fig 9.** (a) Effect of FA, CS, FA–CSNPs, and bionanocomposite on the cell viability of RAW264.7 cells, (b) pinocytic and phagocytic activity of RAW264.7 cells, and (c) fluorescence values, (d–h) Microscopic fluorescence images of (d) control, (e) LPS, (f–h) bionanocomposite (500, 200, and 50 µg mL$^{-1}$) caused phagocytosis. The values are expressed as mean ± standard deviation (n = 3). Significant differences with control cells were calculated as *$P < 0.05$ or **$P < 0.01$.

microscope and provided visual evidence for the measurement of phagocytic uptake. It was observed that the green fluorescence intensity was highest in bionanocomposite film-treated or LPS-treated, followed by FA-CSNPs-treated or LPS-treated in contrast to untreated RAW264.7 cells (Fig 9C). Meanwhile, the FA and CS showed less effect. These results clearly indicated that bionanocomposite film can efficiently encourage the phagocytic abilities of macrophages. Previous reports have shown that the biological properties of the bionanocomposite/nanomaterials were usually dependent on their size, shape, surface area, and components used in the synthesis [82]. The components of FA, CS, and sunflower oil of the bionanocomposite film can be easily bound to mannose receptors to mediate the phagocytosis of macrophages. Thus, the constituents of fly ash and sunflower oil in the bionanocomposite film might be responsible for the phagocytosis enhancement in macrophages.

## Effect on cytokines secretion

The influence of bionanocomposite film on the probable mechanism of cytokines (NO, IL-6, and TNF-α) release in RAW 264.7 cells could be through the activation of macrophages. The basic pathway is based on the secretion of a number of macrophage-derived biological factors such as NO, TNF-α, IL-6, IL-1, IL-10, and IL-12), which are the primary mechanism of immunomodulators [83]. The cytokines NO, IL-6, and TNF- are well-known active chemicals in human beings that play crucial roles in a variety of pathophysiological processes, including host defense mechanisms, inflammation, cancer, and immunological disorders [84]. When the host is invaded by external pathogen threats, activated macrophages may release cytokines to influence the immune system. The current study is concerned with the effect of the bionanocomposite film on the secretion of NO, IL-6, and TNF-α, as well as their secretion mechanism. The results of ELISA indicated that the increase in NO generation of RAW264.7 cells was found concentration-dependent manner (50, 200, and 500 $\mu g\ mL^{-1}$). In contrast to the control group, the quantity of NO was remarkably improved after treatment by 500 $\mu g\ mL^{-1}$ of bionanocomposite film ($P< 0.01$) and reached $50.47\times10^{-3}$ mol $L^{-1}$. This value was higher than the treated cells with 500 $\mu g\ mL^{-1}$ of LPS ($< 4\times10^{-3}$ mol $L^{-1}$) [85]. Similar dose-dependent behavior was observed in the release of IL-6 and TNF-α in RAW264.7 cells after the treatment (Fig 10A–10C).

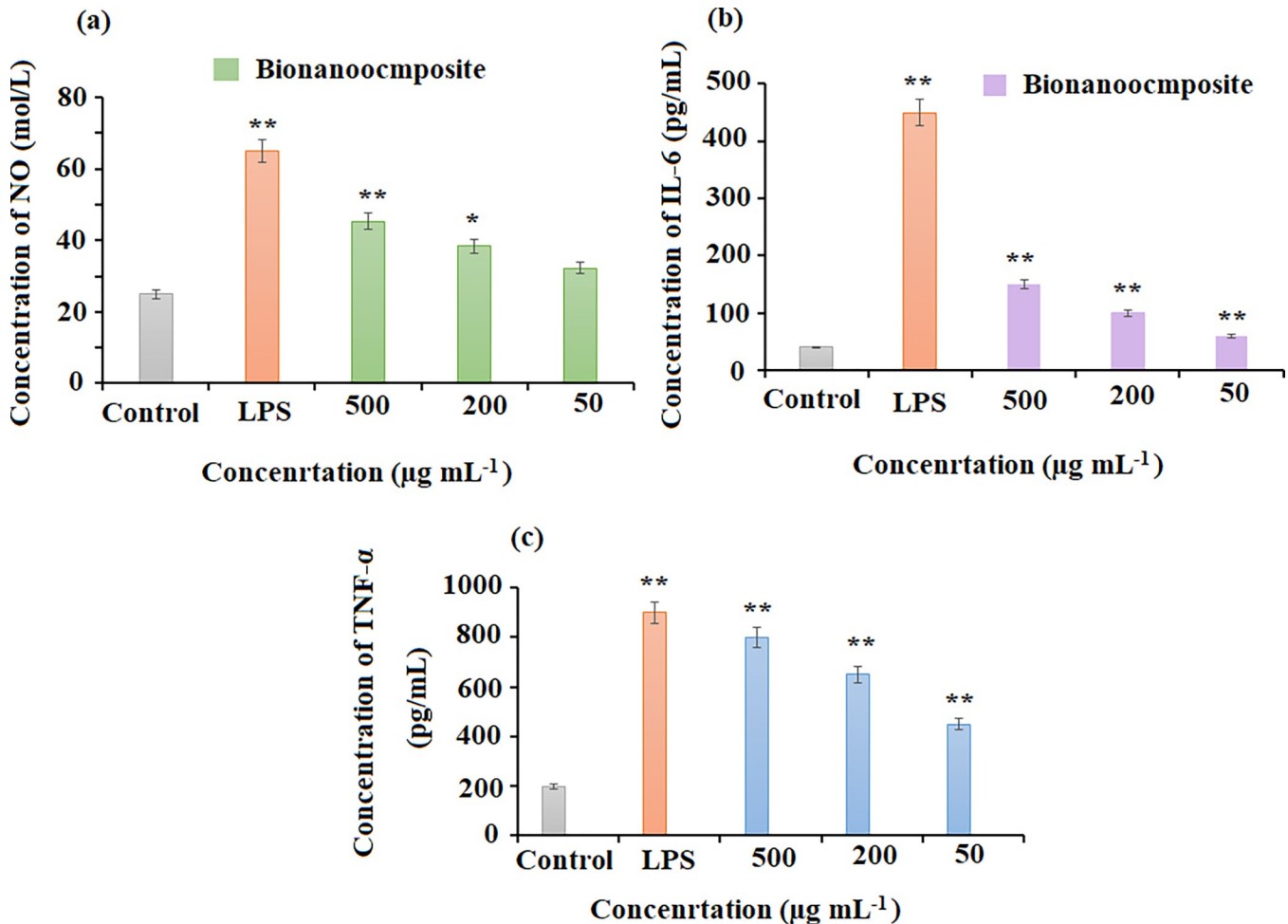

**Fig 10.** Effect of sunflower oil/FA–CSNPs bionanocomposite on the production of cytokines (a) NO, (b) IL–6, and (c) TNF–α. The data are expressed as mean ± standard deviation (n = 3). The significant differences between the treated cells and the control cells were calculated as *$P < 0.05$ or **$P < 0.01$.

The untreated macrophages released a low amount of IL-6 and TNF-α, but when the bionanocomposite film was added, TNF-α and IL-6 levels increased dramatically ($P < 0.01$). The amount of IL-6 treated by the bionanocomposite film (for example, 150 pg mL$^{-1}$ at 500 μg mL$^{-1}$) was smaller than some polysaccharides isolated from *Ganoderma atrum* (1200 pg mL$^{-1}$ at 500 μg mL$^{-1}$ *G. atrum* polysaccharide) [86]. The level of TNF-α was found to be 800 pg mL$^{-1}$ in the treated cells with 500 μg mL$^{-1}$ of bionanocomposite film. These findings presented that the bionanocomposite film showed immunomodulatory properties by enhancing NO, IL-6, and TNF- production in RAW264.7 cells.

## RT-PCR analysis

The RT-PCR was used to examine the gene transcriptions of iNOS, IL-6, and TNF-α, revealing that the rise in NO, IL-6, and TNF-α secretion might be related to transcriptional enhancement [87]. The results demonstrated that RAW264.7 cells treated with the bionanocomposite film or LPS had significantly higher levels of iNOS, IL-6, and TNF-α mRNA transcription which was consistent with NO, IL-6, and TNF-α secretion. According to these findings, the bionanocomposite film enhanced NO, IL-6, and TNF-α production via upregulating the mRNA expression of iNOS, IL-6, and TNF-α (Fig 11A–11C). As previously indicated, the diverse range of cytokines (NO, IL-6, and TNF-α) generated by the bionanocomposite film/

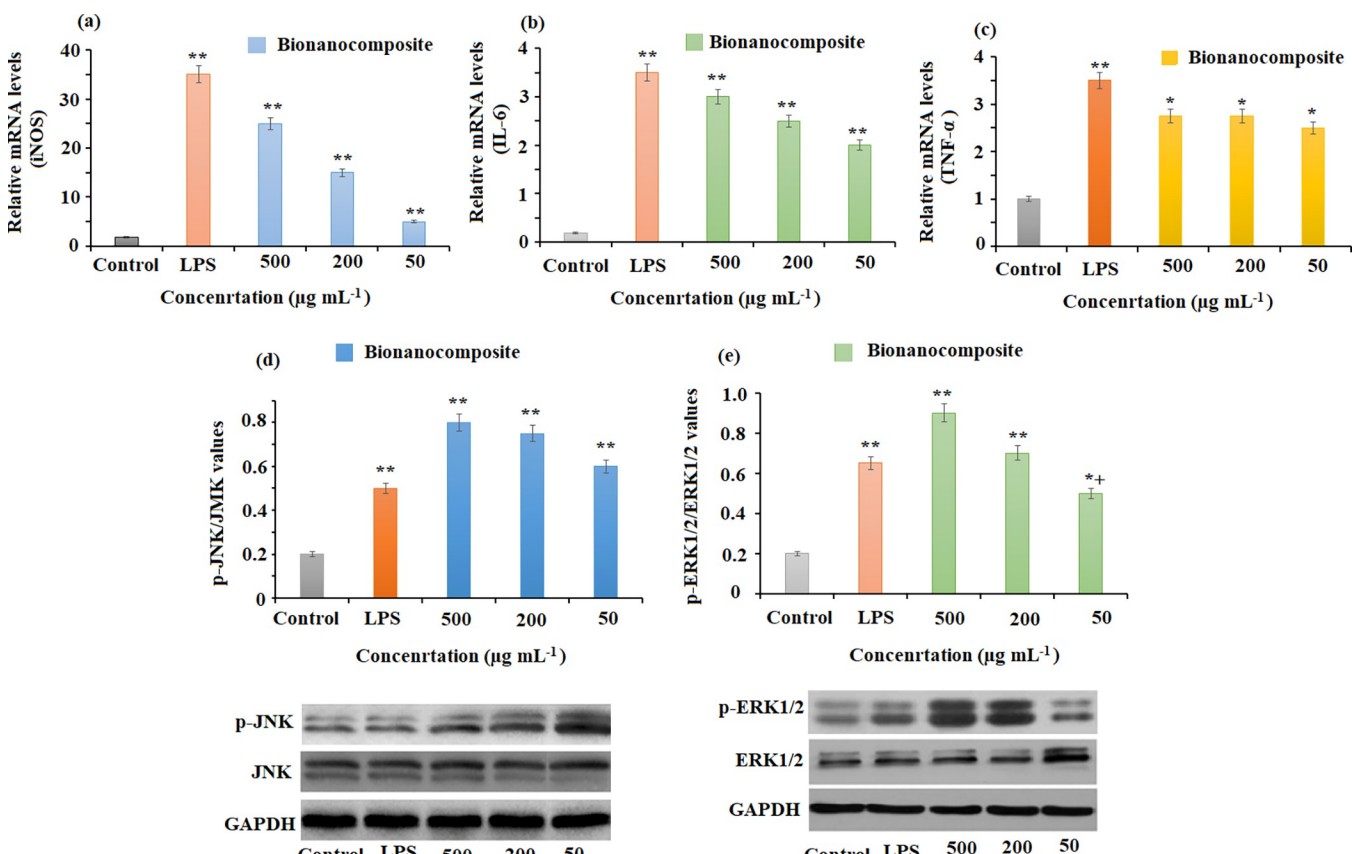

**Fig 11.** (a–c) the expression (iNOS, IL–6, and TNF–α) mRNA in RAW264.7 cells treated with sunflower oil/FA–CSNPs bionanocomposite for 12 h. The values were estimated using $2^{-\Delta\Delta}$ CT method (d and e) the protein expression (p–JNK, JNK, p–ERK1/2 and ERK1/2 treated with the bionanocomposite for 6 h. The protein GAPDH was used as loading control. The results are expressed as mean ± standard deviation (n = 3). Significant differences between the treated cells with the control cells were calculated as *($P < 0.05$) or **($P < 0.01$).

nanomaterials in RAW264.7 cells raises relevant questions regarding the cell signaling activated pathways. Several processes involving cytokine production have been reported, with upregulation of the mitogen-activated protein kinase (MAPK) pathway being one of the most prominent [88]. MAPKs have a role in regulating a variety of cell activities as well as cellular dominance in response to numerous external stimuli. The MAPK family possesses 3 members: p38, JNK, and ERK1/2.

External and internal signals, such as polysaccharides, IL-6, and TNF-α, can activate pathways that lead to gene transcription and cytokine generation [89]. The phosphorylated expression (p-JNK and p-ERK1/2 proteins) in RAW264.7 macrophages were raised in a dose-dependent manner. Furthermore, when the values of p-JNK/total JNK and p-ERK1/2/total ERK 1/2 were compared to the control group, the level of phosphorylation of JNK and ERK1/2 proteins considerably increased ($P < 0.01$). As a result, the bionanocomposite film exhibited a strong pro-phosphorylation effect on JNK and ERK1/2.

## Influence of bionanocomposite film on the production of IL-1β

IL-1β is a well-known cytokine that plays a crucial role in the immune response upper reaches. It can cause the synthesis of a number of inflammatory mediators, which are important for the activation and regulation of immune cells and the inflammatory response [90]. The effect of nanomaterials on the production of IL-1β in RAW264.7 macrophages was evaluated and it was observed that the untreated RAW264.7 cells released small amounts of IL-1β (Fig 12A), while, the addition of bionanocomposite film to the culture media enhanced IL-1β production at concentrations of 200–500 μg mL$^{-1}$ range. The obtained results revealed that the bionanocomposite film has immunomodulatory potential by enhancing the formation of IL-1β, that may be due to macrophage secretion of cytokines (NO, IL-6, and TNF-α). It is essential to

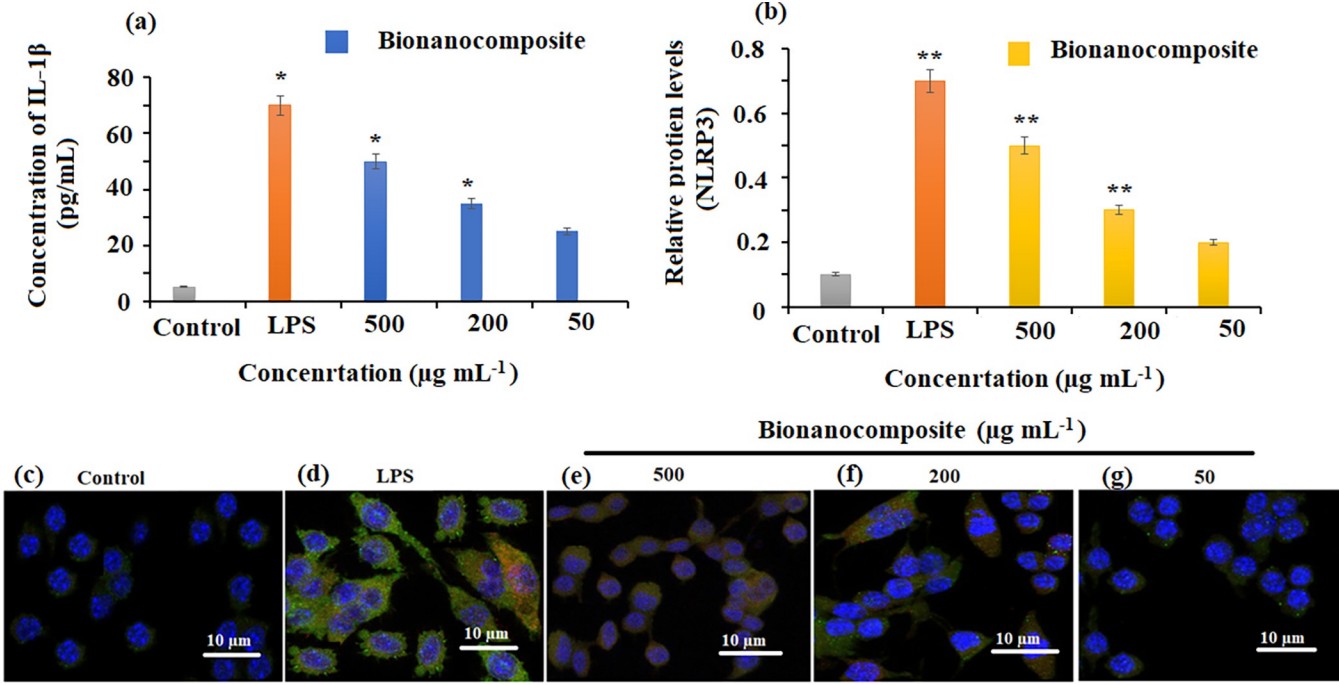

**Fig 12. Effect of sunflower oil/FA–CSNPs bionanocomposite on the production of IL–1β through activating of NLRP3 inflammasome in macrophages.** (a) IL–1β production in RAW 264.7 cells for 24 h. (b) The confocal fluorescence images of NLRP3 and ASC in macrophages. (c–g) the expression of NLRP3 antibody in RAW264.7 cells treated with bionanocomposite for 12 h. The values are calculated as mean ± standard deviation (n = 3). Significant differences with control cells were expressed as *P < 0.05 or **P < 0.01.

explore the molecular mechanism of this activity. Previously, the reports have addressed that the polysaccharides can reduce the generation of IL-1β in LPS-induced macrophages by reducing NF-B activation and phosphorylation of ERK and JNK [91]. However, the IL-1β and NOD-like receptor 3 (NLRP3) inflammasomes have been shown to have a close interaction, and they are thought to serve a key role in the etiology of numerous disorders. The NLRP3 inflammasome can trigger the production of a range of inflammatory mediators in immune cells by producing IL-1β.

This could provide a new target for treating a variety of disorders. The role of the NLRP3 inflammasome in the release of IL-1β by the bionanocomposite film-induced macrophages was studied. NOD-like receptor 3 (NLRP3), the adaptor protein ASC, and caspase-1 make up the NLRP3 inflammasomes, which are found in the cytoplasm [92]. NLRP3 colocalization with ASC was first observed using immunofluorescence labeling. Components of the NLRP3 inflammasome are notoriously difficult to detect in resting cells. The images revealed that NLRP3 expression with ASC was low in the control group, meanwhile NLRP3 colocalization with ASC was clearly elevated in the LPS and the bionanocomposite film groups (Fig 12B). Following that, Western blotting was used to identify the expression of the NLRP3 protein. In a dose-dependent way, NLRP3 protein expression was higher in the bionanocomposite film group than in the control group (Fig 12C–12G). The obtained results indicated that sunflower oil/FA-CSNPs bionanocomposite film has enhanced the production of IL-1β via NLRP3 inflammasome production and activation in RAW264.7 cells.

## Possible pathway of bionanocomposite-induced immunomodulation

The use of nanoparticles (NPs) in medicine showed promising applications. The immune system is a crucial defense mechanism for protecting organisms from external threats. NPs influence the operation of immune system and interact with it, either stimulating or suppressing the immunological response. These moderating effects could be advantageous or harmful. These immunomodulations are influenced by their compositions, sizes, surface chemistry, and other factors [93]. As an efficient source of energy, bionanocomposite film can tiger the production of ROS by transferring energy to the nearby oxygen compound. The increased generation of ROS by bionanocomposite film can cause damage to multiple organelle and cell death [94]. It is necessary to investigate at the biochemical mechanism underlying this activity. Earlier studies have shown that nanoparticles could suppress the generation of IL-β via inhibiting the phosphorylation of ERK and JNK as well as NF-κB activation in LPS-induced macrophages [95]. But it was discovered that IL-1 and NLRP3 inflammasome has close relationship and play essential role in pathogenesis of various diseases. It has been demonstrated that the NLRP3 inflammasome can be triggered by the detection of pathogen-associated molecular patterns (PAMPs) or host-derived signals via intracellular pattern recognition receptor NLR, which causes the cysteine protease caspase-1 recruitment and activation, and subsequent release of IL-18 or IL-1β mature cytokine forms for the protection of host from many viral, bacterial infections and tumors [96]. Moreover, the formation of IL-1β via NLRP3 inflammasome can initiate the generation of diverse inflammatory mediators in immune cells. This could provide many new diseases a new target for the treatment. Furthermore, the bionanocomposite may promote phagocytosis via the mannose receptor (MR). The outcomes showed that the bionanocomposite binds to a variety of receptors and triggers macrophages immunomodulatory response. The probable molecular mechanisms of bionanocomposite-enhanced macrophage immunomodulation were assumed to be principally via two signaling pathways, namely the JNK/ERK MAPKs pathways and the NLRP3 inflammasome signaling pathway, which could give innovative strategies for the curing of several disorders (Scheme 4).

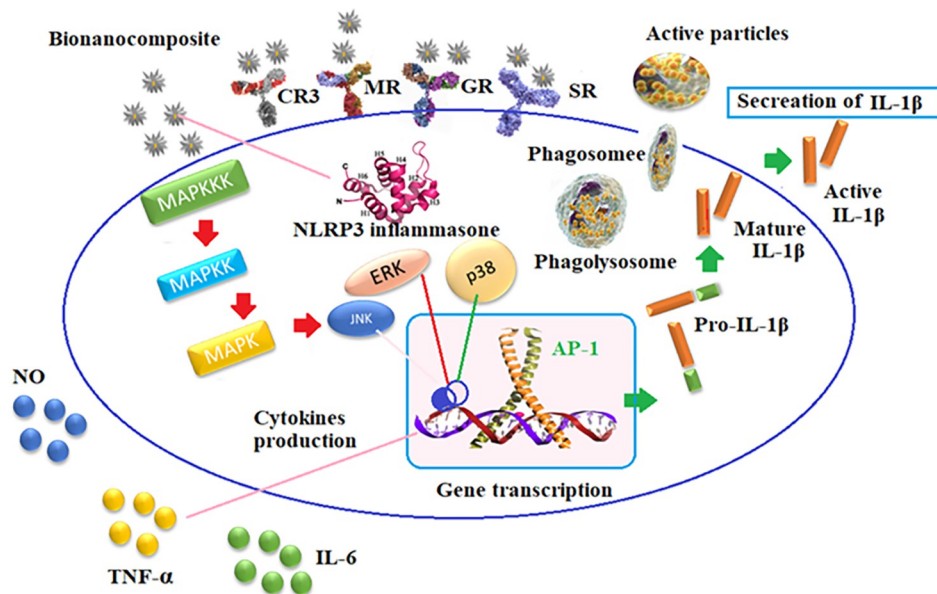

**Scheme 4. Schematic illustration for the effect of bionanocomposite film on the secretion of NO, IL–6, and TNF–α, as well as their secretion mechanism.**

## Conclusion

This study described formation of a novel polymeric sunflower oil- FA-CSNPs bionanocomposite film, using chitosan nanoparticles in the presence of sunflower oil decorated with fly ash. The fabricated bionanocomposite film was characterized and identified by various spectroscopic and microscopic investigations including FTIR, XRD, and SEM confirmed the formation of sunflower oil/FA-CSNPs bionanocomposite film. The average particle size in nanofilm was 100 nm. The FTIR showed the functional groups present in the prepared film as well as the initial components FA, CS, and FA-CSNPs. Excellent antibacterial and immunomodulatory activities were observed. The bionanocomposite showed one wide inhibition zones against *P. aeruginosa* (34 mm) and *B. subtilis* (28 mm) at 100 μg mL$^{-1}$, respectively. The immunomodulatory effect of the pre-synthesized bionanocomposite was tested towards RAW264.7 cells and it was found that the developed bionanocomposite film possessed notable immunomodulatory potential by promoting phagocytosis and enhancing the production of cytokines (NO, IL-6, IL-1β, and TNF-α). The signaling immunomodulation mechanism was conducted through the JNK/ERK MAPKs pathway and the NLRP3 inflammasome signaling pathway, which could give promising strategies for the treatment of several disorders in the future.

## Supporting information

**S1 File.**
(DOCX)

**S1 Raw images.**
(PDF)

## Author Contributions

**Data curation:** Maha F. El-Tohamy.

**Formal analysis:** Seham S. Alterary.

**Investigation:** Maha F. El-Tohamy.

**Methodology:** Musarat Amina.

**Validation:** Seham S. Alterary.

**Visualization:** Seham S. Alterary.

**Writing – original draft:** Maha F. El-Tohamy.

**Writing – review & editing:** Musarat Amina.

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
