## [Decision Letter · Decision Letter 0]

10 Nov 2022

PONE-D-22-25657Biogenic sunflower oil-chitosan decorated fly ash nanocomposite film using white shrimp shell waste: antibacterial and immunomodulatory potentialPLOS ONE

Dear Dr. Amina,

Thank you for submitting your manuscript to PLOS ONE. After careful consideration, we feel that it has merit but does not fully meet PLOS ONE’s publication criteria as it currently stands. Therefore, we invite you to submit a revised version of the manuscript that addresses the points raised during the review process.

We look forward to receiving your revised manuscript.

Kind regards,

Sivasankar Koppala

Academic Editor

PLOS ONE

Journal Requirements:

  "This research was funded by Researchers Supporting Project in King Saud University and the code number is (RSP-2021/195)."  

   "The authors are highly grateful to Researchers Supporting Project number (RSP-2021/195), King Saud University, Riyadh, Saudi Arabia"

 "This research was funded by Researchers Supporting Project in King Saud University and the code number is (RSP-2021/195)."

   "No"

Reviewers' comments:

Reviewer's Responses to Questions

**Comments to the Author**

1. Is the manuscript technically sound, and do the data support the conclusions?

Reviewer #1: Yes

Reviewer #2: Partly

2. Has the statistical analysis been performed appropriately and rigorously? 

Reviewer #1: Yes

Reviewer #2: N/A

3. Have the authors made all data underlying the findings in their manuscript fully available?

Reviewer #1: Yes

Reviewer #2: Yes

4. Is the manuscript presented in an intelligible fashion and written in standard English?

Reviewer #1: Yes

Reviewer #2: Yes

5. Review Comments to the Author

Reviewer #1: Authors have taken up an important topic for their research, the article is well supported by experimental procedures and characterization. However, there are few points which needs to be addressed. My concerns are as follows.

1. The abstract should be precise and more informative.

2. Introduction 3rd paragraphs... Another study conducted by Mitiko. Researcher’s name is underlined for any specific reason or it is a typo?

3. Authors mentioned two reports on chitosan loaded fly ash nanocomposite, they should mention also mention how their procedure are different from previously reported one.

4. If possible introduction should be shortened and the references be reduced.

5. Section: Isolation of chitin and formation of chitosan: Authors developed their own demineralization step or adopted previously reported procedure?

6. Section: Isolation of chitin and formation of chitosan: what is the purpose of bleaching the samples?

7. Section: Isolation of chitin and formation of chitosan: Is there any other reported procedure for deproteinization ?

8. What is the significance of Cell viability assay?

9. “RT-PCR” full form should be mentioned at first place of appearance.

Reviewer #2: Manuscript ID: PONE-D-22-25657

Title: Biogenic sunflower oil-chitosan decorated fly ash nanocomposite film using white shrimp shell waste: antibacterial and immunomodulatory potential

Journal: PLOS ONE

A new sunflower oil-chitosan decorated fly ash (sunflower oil/FA-CSNPs) bionanocomposite film was synthesized using the extract of Litopenaeus vannamei (White shrimp) and monitoring as an antibacterial and immunomodulatory agent. The chitosan (CS) was prepared by deacetylation of chitin isolated from white shrimp extract. Fly ash-chitosan nanoparticles were produced by using glacial acetic acid and sodium tripolyphosphate solution as cross-linkage. The ultrafine polymeric sunflower oil-CS film was fabricated by dissolving CS in glacial acetic acid for 24 h with continuous stirring. The nanostructure of the fabricated polymeric film was confirmed and characterized by different microscopic and spectroscopic approaches including, FTIR, XRD, and SEM. The results are well supported by the conclusion. I recommend major revision of the manuscript before it can be accepted. I request the authors at addressing all comments and suggestions listed below.

Comments and suggestions:

1. “The formed sunflower oil/FA-CSNPs bionanocomposite film showed promising antibacterial and immunomodulatory potentials towards Bacillus subtilis, Pseudomonas aeruginosa and macrophage-derived RAW264.7 cell line, respectively” – Mention the results here.

2. “Keywords: Litopenaeus vannamei; chitin, chitosan; fly ash; antibacterial potential, immunomodulatory effect."- Use different keywords, which are not used in title.

3. “Recently, the control of organic waste disposal is a growing issue

for the legislation and environmental safety..”--- I suggest the author, to discuss a paragraph related to pollution due to the presence of different contaminants. The authors are recommended to check the below related references, which will improve the supporting information.

Journal of Molecular Liquids 317, 2020, 113916

Separation Science and Technology 55 (10), 2020, 1766-1775

Journal of Cleaner Production 156, 2017, 426-436

Journal of Hazardous Materials 400, 2020, 123247

4. Also in recent times, there has been a great incentive for generating environmentally friendly materials for water purification and wastewater treatment, which need to be included in introduction section.

Photochemistry and photobiology 94 (5), 2018, 935-941

Nanomaterials 9 (5), 2019, 776

Arabian Journal of Chemistry 12 (5), 2019, 633-651

Environmental research 2019, 170, 389-397

5. “White shrimp (L. vannamei) shells were purchased from a local market in Riyadh, Saudi

Arabia.”-what basic criterion were chosen for shrimp collections?

6. “Figure 1: FTIR spectra of (a) Fly ash, (b) chitosan, (c) FA-CSNPs, and (d) Sunflower oil/FACSNPs-bionanocomposite film measured at wavenumber 4400-400 cm-1”. mention the optimal conditions in each figure, while needed.

7 “The dried homogenized shrimp shells (100 g) were demineralized (removing calcium carbonate

and phosphate) by adding 1.0 mol L-1 hydrochloric acid (1L) at room temperature under

agitation for 12 h at 250 rpm.”- Does the author followed any reported work, if yes then the novelty needs to be discussed?

8. “The FTIR spectrum of raw fly ash exhibited broad absorption peaks at 3449 and 1634 cm-1 assigned to O-H and H-O-H groups stretching and deformation vibrations of the water molecules” - Need supporting information, Journal of environmental management 219, 2018, 285-293

9. “The crystalline size of FA-CSNPs was found to be 6.90 nm when calculated using Scherrer’s equation:” - Desalination and Water Treatment 57 (46), 2016, 21863-21869

10. “The impact of FA, FA-CSNPs, and bionanocomposite film on the shape of both treated and untreated B. subtilis and P. aeruginosa were examined using a scanning electron microscope (SEM).”- Provide the details of the manufacturer and also do same for all instrument used including city and country in experimental section.

11. The English quality not up to the mark. All the typos and grammar need to check thoroughly in the manuscript.

6. PLOS authors have the option to publish the peer review history of their article (what does this mean?). If published, this will include your full peer review and any attached files.

Reviewer #1: No

Reviewer #2: No

---

## [Author Response · Author response to Decision Letter 0]

6 Jan 2023

Response letter

PLOS ONE

PONE-D-22-25657

Manuscript Title: Biogenic sunflower oil-chitosan decorated fly ash nanocomposite film using white shrimp shell waste: antibacterial and immunomodulatory potential

https://journals.plos.org/plosone/s/file?id=wjVg/PLOSOne_formatting_sample_main_body.pdfand
https://journals.plos.org/plosone/s/file?id=ba62/PLOSOne_formatting_sample_title_authors_affiliations.pdf

Answer: The manuscript completely meets PLOS ONE style format requirements.

Thank you for stating the following financial disclosure: 

"This research was funded by Researchers Supporting Project in King Saud University and the code number is (RSP-2021/195).

 Answer: I would like to change the financial disclosure as “The authors extend their appreciation to the Deputyship for Research & Innovation, Ministry of Education in Saudi Arabia for funding this research work through the project no. (IFKSURG-2-000)” as the above stated financial project has been finished. 

Answer: The funder has full participation in the manuscript beside financial support. The author participated in visualization, formal analysis, and validation.

Answer: I have replied the query. 

Thank you for stating the following in your Competing Interests section: 

 "No"

Competing Interests: The authors have declared that no competing interests exist.

In your Data Availability statement, you have not specified where the minimal data set underlying the results described in your manuscript can be found. PLOS defines a study's minimal data set as the underlying data used to reach the conclusions drawn in the manuscript and any additional data required to replicate the reported study findings in their entirety. All PLOS journals require that the minimal data set be made fully available. For more information about our data policy, please see http://journals.plos.org/plosone/s/data-availability.

Answer: Not applicable.

Reviewer #1:

Thank You so much for your valuable comments. We appreciate your supporting comments as it helped us to improve this manuscript. All comments have been considered in the revised manuscript and all corrections have been highlighted using red color. 

1. The abstract should be precise and more informative.

Answer: The abstract has been modified, concise, and more information have been included. 

2. Introduction 3rd paragraphs... Another study conducted by Mitiko. Researcher’s name is underlined for any specific reason or it is a typo?

Answer: The underline of the researcher’s name has been removed. it was a typo graphical error.

3. Authors mentioned two reports on chitosan loaded fly ash nanocomposite, they should mention also mention how their procedure are different from previously reported one.

Answer: Our study report for the first time the preparation of bionanocomposite film using sunflower oil with fly ash decorated with chitosan nanoparticles which is different from previously published reports. The novelty of the study has been incorporated in the text and highlighted

4. If possible, introduction should be shortened and the references be reduced.

Answer: The introduction section has been shortened as suggested and the references have been minimized.

5. Section: Isolation of chitin and formation of chitosan: Authors developed their own demineralization step or adopted previously reported procedure?

Answer: The isolation and chitosan formation were obeyed (de Queiroz Antonino etal 2017) and it has been cited in the revised text (section . 

de Queiroz Antonino RS, Lia Fook BR, de Oliveira Lima VA, de Farias Rached RÍ, Lima EP, da Silva Lima RJ, Peniche Covas CA, Lia Fook MV. Preparation and characterization of chitosan obtained from shells of shrimp (Litopenaeus vannamei Boone). Marine drugs. 2017 May 15;15(5):141.

6. Section: Isolation of chitin and formation of chitosan: what is the purpose of bleaching the samples?

Answer: To obtain the pure crystalline form of chitin and chitosan.

7. Section: Isolation of chitin and formation of chitosan: Is there any other reported procedure for deproteinization? 

Answer: There are few methods reported in the literature for the deproteinization of chitin and formation of chitosan which includes, ultrasound deproteinization of chitin, enzymatic deproteinization and demineralization using natural deep eutectic solvents for production of insect chitin from shrimp head waste and most common procedures is enzymatic deproteinization. But we applied more simple chemical demineralization and deproteinization process [ref]. The above mentioned statement has been included in the text along with the references.

- Vallejo-Domínguez D, Rubio-Rosas E, Aguila-Almanza E, Hernández-Cocoletzi H, Ramos-Cassellis ME, Luna-Guevara ML, Rambabu K, Manickam S, Munawaroh HS, Show PL. Ultrasound in the deproteinization process for chitin and chitosan production. Ultrasonics Sonochemistry. 2021 Apr 1;72:105417. https://doi.org/10.1016/j.ultsonch.2020.105417

- Zhou P, Li J, Yan T, Wang X, Huang J, Kuang Z, Ye M, Pan M. Selectivity of deproteinization and demineralization using natural deep eutectic solvents for production of insect chitin (Hermetia illucens). Carbohydrate polymers. 2019 Dec 1;225:115255. https://doi.org/10.1016/j.carbpol.2019.115255

- Younes I, Ghorbel-Bellaaj O, Nasri R, Chaabouni M, Rinaudo M, Nasri M. Chitin and chitosan preparation from shrimp shells using optimized enzymatic deproteinization. Process Biochemistry. 2012 Dec 1;47(12):2032-9. https://doi.org/10.1016/j.procbio.2012.07.017

- de Queiroz Antonino RS, Lia Fook BR, de Oliveira Lima VA, de Farias Rached RÍ, Lima EP, da Silva Lima RJ, Peniche Covas CA, Lia Fook MV. Preparation and characterization of chitosan obtained from shells of shrimp (Litopenaeus vannamei Boone). Marine drugs. 2017 May 15;15(5):141. https://doi.org/10.3390/md15050141

8. What is the significance of Cell viability assay?

Answer: Cell viability is a measure of the proportion of live cells within a population. The measurement of cell viability plays an essential role in all forms of cell culture. The significant has been incorporated in the cell viability assay section in the text.

9. “RT-PCR” full form should be mentioned at first place of appearance.

Answer: The full form of RT-PCR (Real time-polymerase chain reaction) has been included in the text. 

Response letter

Journal PLOS ONE

PONE-D-22-25657

Manuscript Title: Biogenic sunflower oil-chitosan decorated fly ash nanocomposite film using white shrimp shell waste: antibacterial and immunomodulatory potential

Reviewer #2:

Thank You so much for your valuable comments. We appreciate your supporting comments as it helped us to improve this manuscript. All comments have been considered in the revised manuscript and all corrections have been highlighted using red color. 

1. The formed sunflower oil/FA-CSNPs bionanocomposite film showed promising antibacterial and immunomodulatory potentials towards Bacillus subtilis, Pseudomonas aeruginosa and macrophage-derived RAW264.7 cell line, respectively” – Mention the results here.

Answer: After this statement the results has been included in the abstract with their zone of inhibition for each bacterial strain and optimal active concentration of macrophage-derived RAW264.7 cell line. The results have been highlighted in the text.

2. Keywords: Litopenaeus vannamei; chitin, chitosan; fly ash; antibacterial potential, immunomodulatory effect."- Use different keywords, which are not used in title.Answer: The keywords have been changed as your suggestion.

3. “Recently, the control of organic waste disposal is a growing issue

for the legislation and environmental safety..”--- I suggest the author, to discuss a paragraph related to pollution due to the presence of different contaminants. The authors are recommended to check the below related references, which will improve the supporting information.

Journal of Molecular Liquids 317, 2020, 113916

Separation Science and Technology 55 (10), 2020, 1766-1775

Journal of Cleaner Production 156, 2017, 426-436

Journal of Hazardous Materials 400, 2020, 123247

Answer: The paragraph discussing the pollution created by different contaminants has been included in the text along with the citation of suggested references.

4. Also in recent times, there has been a great incentive for generating environmentally friendly materials for water purification and wastewater treatment, which need to be included in introduction section.

Photochemistry and photobiology 94 (5), 2018, 935-941

Nanomaterials 9 (5), 2019, 776

Arabian Journal of Chemistry 12 (5), 2019, 633-651

Environmental research 2019, 170, 389-397

Answer: The paragraph also mentioned water waste treatment by environmentally friendly materials and the suggested references has been cited in the text 

5. “White shrimp (L. vannamei) shells were purchased from a local market in Riyadh, Saudi

Arabia.”-what basic criterion were chosen for shrimp collections?

Answer: Fully mature shrimps with well grown shells were purchased from the markets and cleaned with water to remove the debris. The selection criteria have been included in the text in the materials and method section. 

6. “Figure 1: FTIR spectra of (a) Fly ash, (b) chitosan, (c) FA-CSNPs, and (d) Sunflower oil/FACSNPs-bionanocomposite film measured at wavenumber 4400-400 cm-1”. mention the optimal conditions in each figure, while needed.

Answer: The optimal condition used for FTIR have been incorporated in the caption of the FTIR figure and highlighted using red color.

7. The dried homogenized shrimp shells (100 g) were demineralized (removing calcium carbonate and phosphate) by adding 1.0 mol L-1 hydrochloric acid (1L) at room temperature under agitation for 12 h at 250 rpm.”- Does the author followed any reported work, if yes then the novelty needs to be discussed? 

Answer: The process of demineralized and deproteination have been obeyed (de Queiroz Antonino et al 2017). With slight modification and the reference is cited in the text. However, our study report for the first time the preparation of bionanocomposite film using sunflower oil with fly ash decorated with chitosan nanoparticles which is different from previously published reports. The novelty of the study has been incorporated in the text and highlighted. 

8. “The FTIR spectrum of raw fly ash exhibited broad absorption peaks at 3449 and 1634 cm-1 assigned to O-H and H-O-H groups stretching and deformation vibrations of the water molecules” - Need supporting information, Journal of environmental management 219, 2018, 285-293

Kenawy ER, Ghfar AA, Wabaidur SM, Khan MA, Siddiqui MR, Alothman ZA, Alqadami AA, Hamid M. Cetyltrimethylammonium bromide intercalated and branched polyhydroxystyrene functionalized montmorillonite clay to sequester cationic dyes. Journal of Environmental Management. 2018 Aug 1;219:285-93. https://doi.org/10.1016/j.jenvman.2018.04.121

Answer: The reference has been added in the text and the reference section.

9. “The crystalline size of FA-CSNPs was found to be 6.90 nm when calculated using Scherrer’s equation:” - Desalination and Water Treatment 57 (46), 2016, 21863-21869

Answer: The suggested reference has been included in the text and in the reference section.

Mittal A, Naushad M, Sharma G, Alothman ZA, Wabaidur SM, Alam M. Fabrication of MWCNTs/ThO2 nanocomposite and its adsorption behavior for the removal of Pb (II) metal from aqueous medium. Desalination and Water Treatment. 2016 Oct 1;57(46):21863-9. https://doi.org/10.1080/19443994.2015.1125805

10. “The impact of FA, FA-CSNPs, and bionanocomposite film on the shape of both treated and untreated B. subtilis and P. aeruginosa were examined using a scanning electron microscope (SEM).”- Provide the details of the manufacturer and also do same for all instrument used including city and country in experimental section.

Answer: The manufacturer’s details including city and country for scanning electron microscopy as well as all the instruments have been incorporated in the experimental section. And highlighted using red color. 

11. The English quality not up to the mark. All the typos and grammar need to check thoroughly in the manuscript.

Answer: The manuscript has been revised carefully and all typos and grammar have been revised and improved.

---

## [Decision Letter · Decision Letter 1]

22 Feb 2023

Biogenic sunflower oil-chitosan decorated fly ash nanocomposite film using white shrimp shell waste: antibacterial and immunomodulatory potential

PONE-D-22-25657R1

Dear Dr. Amina,

We’re pleased to inform you that your manuscript has been judged scientifically suitable for publication and will be formally accepted for publication once it meets all outstanding technical requirements.

Kind regards,

Sivasankar Koppala

Academic Editor

PLOS ONE

Additional Editor Comments (optional):

Reviewers' comments:

Reviewer's Responses to Questions

**Comments to the Author**

1. If the authors have adequately addressed your comments raised in a previous round of review and you feel that this manuscript is now acceptable for publication, you may indicate that here to bypass the “Comments to the Author” section, enter your conflict of interest statement in the “Confidential to Editor” section, and submit your "Accept" recommendation.

Reviewer #1: All comments have been addressed

Reviewer #2: All comments have been addressed

2. Is the manuscript technically sound, and do the data support the conclusions?

Reviewer #1: Yes

Reviewer #2: Yes

3. Has the statistical analysis been performed appropriately and rigorously? 

Reviewer #1: N/A

Reviewer #2: Yes

4. Have the authors made all data underlying the findings in their manuscript fully available?

Reviewer #1: Yes

Reviewer #2: Yes

5. Is the manuscript presented in an intelligible fashion and written in standard English?

Reviewer #1: Yes

Reviewer #2: Yes

6. Review Comments to the Author

Reviewer #1: Authors have made all the required corrections satisfactorily.

Reviewer #2: All the comments raised by me are addressed properly. I suggest acceptance of the paper in the current form.

7. PLOS authors have the option to publish the peer review history of their article (what does this mean?). If published, this will include your full peer review and any attached files.

Reviewer #1: No

Reviewer #2: No

---

## [Editor Report · Acceptance letter]

23 Mar 2023

PONE-D-22-25657R1 

Biogenic sunflower oil-chitosan decorated fly ash nanocomposite film using white shrimp shell waste: antibacterial and immunomodulatory potential 

Dear Dr. Amina:

I'm pleased to inform you that your manuscript has been deemed suitable for publication in PLOS ONE. Congratulations! Your manuscript is now with our production department. 

Kind regards, 

on behalf of

Dr. Sivasankar Koppala 

Academic Editor

PLOS ONE